# A Combined Fractional Order Repetitive Controller and Dynamic Gain Regulator for Speed Ripple Suppression in PMSM Drives

Haohao Guo [1], Fengkui Zhang [1], Qiaofen Zhang [2], Yancheng Liu [1,*], Tianxiang Xiang [1] and Jintong Xing [1]

[1] School of Marine Engineering, Dalian Maritime University, Dalian 116026, China; ghh1984@dlmu.edu.cn (H.G.); zhangfengkui@dlmu.edu.cn (F.Z.); xjt0912@dlmu.edu.cn (J.X.)
[2] School of Navigation and Naval Architecture, Dalian Ocean University, Dalian 116023, China; zqf@dlou.edu.cn
* Correspondence: liuyc@dlmu.edu.cn

**Abstract:** Repetitive control (RC) has been widely used in many fields due to its excellent ability to suppress periodic disturbances. However, when the permanent magnet synchronous motor (PMSM) operates at variable speeds, the speed loop sampling frequency is usually not equal to an integer multiple of the fundamental frequency of speed ripple, which prevents disturbances from being completely suppressed. In addition, the open-loop gain of the motor control system with RC is too large at certain frequencies, resulting in excessive speed overshoot during startup and loading. To solve these two problems, this paper proposes a fractional order repetitive control (FORC) strategy with dynamically adjustable gain. A fractional order delay link is introduced to make up for the shortcomings of the conventional repetitive controller (CRC) in its ability to suppress periodic speed ripples when the sampling frequency is not an integer multiple of the fundamental frequency of the motor. Then, to weaken the speed overshoot caused by RC, a nonlinear function $fal(e,\alpha,\delta)$ is added in the front of the FORC to dynamically adjust the FORC gain. Simulation and experimental results verify the effectiveness of the proposed method.

**Keywords:** permanent magnet synchronous motors (PMSMs); fractional order repetitive control; speed ripple suppression; overshoot peaks; adaptive gain adjustment

## 1. Introduction

In recent decades, permanent magnet synchronous motors (PMSMs) have been widely used in many fields, such as precision machine tools, electric propulsion ships, automobiles, and household appliances [1–3]. Compared to induction motors, PMSMs are favored for high-performance servo applications due to their high efficiency, high power density, and large torque-to-inertia ratio. However, the PMSM vector control system has a series of non-ideal factors, such as current measurement error, a dead zone effect, cogging torque, and flux harmonics [4]. These non-ideal factors aggravate steady-state torque/speed fluctuations and increase the harmonic current content in the motor, which is unacceptable in applications requiring a speed-ripple-free performance in the motor [5]. In electric vehicle applications, the cogging torque and flux harmonics cause low-frequency speed fluctuations, which can make passengers uncomfortable [6]. The torsional vibrations in elastic drive systems caused by speed fluctuations can lead to shafting cracking and/or fatigue cycles [7]. Therefore, it is necessary to suppress the speed fluctuations in PMSMs caused by non-ideal factors.

Many effective methods have been proposed in the literature to suppress the adverse impact of these non-ideal factors on the control performance of PMSMs [8–13]. In [8], an adaptive linear neuron-based dead time compensation method is proposed for vector-controlled PMSM drives, but with a low response at low speeds and without a design

standard for learning rate. Reference [9] proposes a self-calibration strategy for the phase current sensor, but this method needs to change the standard connection of the current sensor, and it can only be used in driving systems using a Hall effect current sensor. In [10], an offset error compensation method that considers the influence of the outer loop controllers is applied to improve the motor's dynamic performance. However, this method may fail when the current controller is saturated. In references [11–13], parallel resonant controllers and PI controllers are used to suppress the periodic speed fluctuations in the PMSM caused by the current measurement errors, dead zone effect, load disturbance, and other factors, but the multi-resonant controllers increase the computational burden. Reference [14] discusses the negative effect of current measurement errors from the perspective of flux linkage estimation and uses the current loop output results to construct error observers. However, this method is limited by the motor speed and load in the practical application. Reference [15] proposes a robust iterative learning control strategy with sliding mode control and iterative learning control to suppress the torque ripple in a PMSM.

The essential reason why current measurement error causes steady-state speed ripples in PMSMs is that the *q*-axis steady-state current is not a constant. It includes the harmonic components related to the electrical angular frequency, which can be seen as a periodic disturbance in the forward channel of the speed loop [11]. Other non-ideal factors such as the cogging torque, dead time effect, and flux harmonics also bring about periodic torque disturbances. To suppress these periodic disturbances, repetitive controllers are widely used for specific harmonic suppression because they can increase the gains of the control system at the disturbance frequencies [16–19]. In [16,17], a modified repetitive controller is inserted into the current inner loop to reduce the sixth-order current harmonics caused by the non-sinusoidal back EMF and dead time. However, this type of method cannot suppress the influence of non-ideal factors such as speed loop forward channel disturbances and current measurement errors. Reference [18] proposes a torque ripple reduction strategy for PMSMs that combines angle-based RC and deadbeat current control. However, the principle of parameter selection to ensure the stability of the system is not given. Reference [19] combines angle-based RC and a disturbance observer into a new type of angle-based repetitive observer for the first time to reduce the torque ripple in PMSMs. Reference [20] reviews the robust control strategies widely used for PMSM speed regulation and advises that future research should prioritize controllers suitable for practical applications and with lower costs, such as SMC, PI, and ESO controllers. In our previous work, a plug-in RC strategy was proposed to suppress the speed ripple caused by current measurement errors [21]. However, this strategy still has the following problems: (1) When the motor speed needs to change continuously, the period of the speed ripple is often not an integer multiple of the speed loop sampling period, which makes the fundamental frequency of the RC deviate from that of the speed ripple, resulting in an incomplete suppression of the speed ripple. (2) The high open-loop gain of the RC results in a large overshoot of the motor speed during the periods of startup and loading, which is unacceptable in many industrial applications.

In this paper, an improved *fal*-FORC strategy is proposed to solve these two problems. Firstly, a fractional order delay link is introduced to make up for the deficiency of the CRC in suppressing the periodic speed ripples at the frequencies of non-integer multiples of the fundamental frequency. Secondly, for the purpose of suppressing the speed overshoot, the nonlinear function *fal*($e,\alpha,\delta$) is added in front of the FORC to adjust the FORC gain dynamically. The paper is organized in the following way. Section 2 analyzes the impact of current measurement error and the shortcomings of the CRC, including poor ripple suppression under variable speeds and excessive overshoot. Section 3 elaborates on the proposed *fal*-FORC strategy and the selection criteria for the related parameters and gives the simulation results. The experimental results are shown in Section 4 to verify the effectiveness of the proposed strategy. Section 5 concludes the paper.

## 2. Current Measurement Error and CRC Defect Analysis

### 2.1. Current Measurement Error

The current measurement channel includes Hall effect current sensors, signal matching circuits, noise filter circuits, AD converters, etc. The equipment tolerance, temperature drift, imbalance of the positive and negative supply voltages of the sensor, aging, noise, etc., of these units may cause bias. The current measurement error includes offset error and scaling error, which cause the steady-state torque and speed of the motor to generate first- and second-order ripples (relative to the motor's fundamental frequency) [3,10], affecting the ride comfort of ships, electric vehicles, and other transportation vehicles.

The research object of this article is a surface-mounted permanent magnet synchronous motor drive system using the $i_d{}^* = 0$ control strategy. A structural block diagram of the permanent magnet synchronous motor vector control system considering the current measurement error is shown in Figure 1, where $k_{sp}$, $k_{si}$, $k_{cp}$, and $k_{ci}$ are the proportional integral coefficients of the speed loop and current loop, respectively; $K_s$ and $T_d$ are the proportional coefficient and delay/time constant of the inverter, respectively; $L_d$, $L_q$, and $R$ are the dq-axis inductance and coil resistance, respectively; $E(s)$ and $K_e$ are the back electromotive force and its coefficient, respectively; $J$ and $B$ are the moment of inertia and the damping coefficient of the motor, respectively; $p$ is the number of motor pole pairs; $\psi_i$ is the permanent magnet flux linkage; $T_L$ is the load torque; $\Delta i_{d\_err}$ and $\Delta i_{q\_err}$ are the disturbances introduced by the current measurement error on the dq-axis, respectively; and $\Delta u_d$ and $\Delta u_q$ are the dq-axis voltage harmonics caused by the dead zone effect.

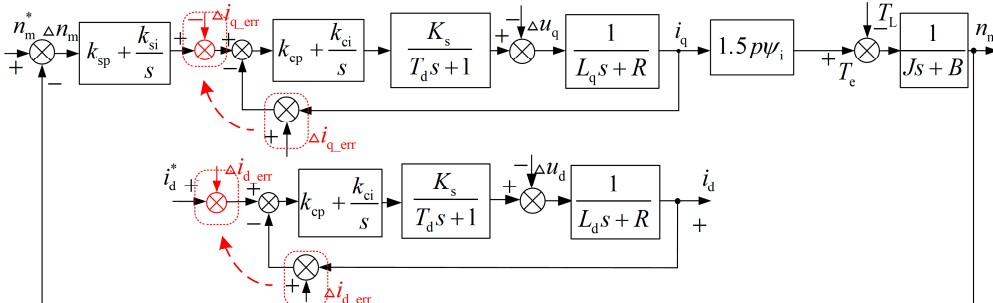

**Figure 1.** Block diagram of PMSM vector control system with current measurement error.

Ignore the effects of back electromotive force and inverter delay and select the PI parameters of the current inner loop to offset the electrical time constant of the motor. The current loop transfer function is equivalent to $G_c(s) = \omega_{cu}/(s + \omega_{cu})$, where $\omega_{cu}$ is the bandwidth of the current loop. The transfer function between the q-axis current measurement error $\Delta i_{q\_err}(s)$ and the rotational speed error $\Delta n_m(s)$ can be obtained as

$$G_{error}(s) = \frac{\Delta n_m(s)}{\Delta i_{q\_err}(s)} = \frac{1.5p\psi_i G_c(s)/(Js+B)}{1+1.5(k_{sp}s+k_{si})p\psi_i G_c(s)/s(Js+B)}, \tag{1}$$

Offset error refers to a certain amount of dc component superimposed on the actual sinusoidal current of the motor, which is mainly caused by zero drift, the residual current in the current sensor, operational amplifier deviation, AD converter deviation, etc. [3]. The q-axis current error [10] caused by the offset error is

$$\Delta i_{q\_offset} = a\cos(\omega_e t + \alpha)\sqrt{\Delta I_{A\_offset}^2 + \Delta I_{A\_offset}\Delta I_{B\_offset} + \Delta I_{B\_offset}^2}, \tag{2}$$

$$\alpha = \tan^{-1}(\sqrt{3}\Delta I_{A\_offset}/(\Delta I_{A\_offset} + 2\Delta I_{B\_offset})), \tag{3}$$

where $a = 2/\sqrt{3}$; $\Delta I_{A\_offset}$ and $\Delta I_{B\_offset}$ are the offset errors of phases A and B, respectively; and $\omega_e$ is the electrical angular frequency of the motor.

(1)    Effect of offset error on motor steady-state speed

Substituting Equation (2) into Equation (1), it can be seen that the steady-state speed deviation $\Delta n_{\text{r\_ss}}^{\text{offset}}(t)$ caused by the offset error can be given by

$$\Delta n_{\text{r\_ss}}^{\text{offset}}(t) = aM_1\cos(\omega_{\text{e}}t + \alpha + \theta_1)\sqrt{\Delta I_{\text{a\_offset}}^2 + \Delta I_{\text{a\_offset}}\Delta I_{\text{b\_offset}} + \Delta I_{\text{b\_offset}}^2}, \quad (4)$$

where $M_1 = |G_{\text{error}}(j\omega_{\text{e}})|$; $\theta_1 = \angle G_{\text{error}}(j\omega_{\text{e}})$. That is, the offset error will cause the first-order ripple in the motor's steady-state speed.

Since the current loop bandwidth is generally much larger than the electrical angular frequency of the motor, the current loop transfer function $G_{\text{c}}(s)$ is approximately 1 at this time, and Equation (1) can be simplified into

$$G_{\text{error}}(s) = \frac{1.5p\psi_{\text{i}}/(Js+B)}{1 + 1.5(k_{\text{sp}}s + k_{\text{si}})p\psi_{\text{i}}/s(Js+B)} = \frac{1.5p\psi_{\text{i}}}{Js + 1.5p\psi_{\text{i}}k_{\text{si}}/s + (1.5p\psi_{\text{i}}k_{\text{sp}} + B)}, \quad (5)$$

The following conclusions can be drawn from the above equations:

① When $\omega_{\text{e}} = \sqrt{1.5p\psi_{\text{i}}k_{\text{si}}/J}$, the maximum steady-state speed ripple value $M_{1\text{max}}$ caused by the offset error is

$$M_{1\text{max}} = \frac{1.5p\psi_{\text{i}}}{2\sqrt{1.5p\psi_{\text{i}}k_{\text{si}}J} + (1.5p\psi_{\text{i}}k_{\text{sp}} + B)}, \quad (6)$$

② The larger the PI parameter of the speed loop, the smaller the speed ripple caused by the offset error. The PI parameter selection of the speed outer loop is generally proportional to the motor's rotational inertia. Therefore, the smaller the rotational inertia, the larger the speed ripple caused by the offset error.

Current sampling includes sensors, operational amplifier conditioning circuits, AD converters, and other units. These processes may cause scaling errors, and the q-axis current error [3] caused by them can be expressed as

$$\Delta i_{\text{q\_scaling}} = \frac{k_b - k_a}{\sqrt{3}}I\cos(2\omega_{\text{e}}t - \frac{\pi}{6}), \quad (7)$$

where $k_a$ and $k_b$ are scaling error coefficients. If they are not equal to 1, that means that there is a scaling error in the current measurement value. Due to the difference between the two sampling conditioning circuits, $k_a \neq k_b$. $I$ is the magnitude of the phase current. Substituting Equation (7) into Equation (1), it can be seen that the steady-state speed deviation $\Delta n_{\text{r\_ss}}^{\text{scaling}}(t)$ caused by the scaling error is

$$\Delta n_{\text{r\_ss}}^{\text{scaling}}(t) = (k_b - k_a)IM_2/\sqrt{3}\cos(2\omega_{\text{e}}t + \frac{\pi}{3} + \theta_2), \quad (8)$$

where $M_2 = |G_{\text{error}}(j2\omega_{\text{e}})|$; $\theta_2 = \angle G_{\text{error}}(j2\omega_{\text{e}})$, that is, the scaling error will cause the second-order ripple in the motor's steady-state speed. The influence of the scaling error is similar to that of the offset error on the steady-state rotation speed at different frequencies and will not be described again.

## 2.2. Unsatisfactory Ripple Suppression under Variable Speeds

The current measurement error and other non-ideal factors can cause 1st-, 2nd-, 3rd-, 4th-... $n$th-order speed fluctuating components, which can be well suppressed using a strategy with a plug-in repetitive controller [21]. The structure of the CRC is shown in Figure 2, where $k_{\text{rc}}$ is the gain in the RC, $z^{-N}$ is the delay link, and $N$ is the ratio of the fundamental period of the disturbance signal to the sampling period of the speed loop. $Q(z)$

is a low-pass filter, and $C(z)$ is a phase compensator. The parameter determination criteria of the CRC are given in detail in [21]. The transfer function of the CRC can be expressed as

$$G_{\mathrm{RC}}(z) = \frac{k_{\mathrm{rc}}C(z)Q(z)z^{-N}}{1 - Q(z)z^{-N}}, \tag{9}$$

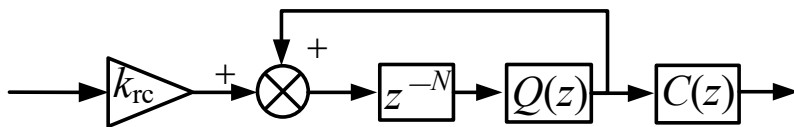

**Figure 2.** The structure of CRC.

With the PMSM parameters given in Table 1, when the motor speed $n_{\mathrm{m}}$ is 300 [rpm], the fundamental frequency of the disturbance signal $f_0$ is 20 Hz. And the speed loop sampling frequency $f_{\mathrm{s}}$ is 1000 Hz. The open-loop amplitude–frequency characteristic of the CRC is shown in Figure 3, where $N = f_{\mathrm{s}}/f_0 = 50$.

**Table 1.** Parameters of PMSM.

| Parameters | Value |
|---|---|
| Rated power $P_{\mathrm{N}}$/W | 88 |
| Rated torque $T_{\mathrm{N}}$/[N·m] | 0.23 |
| Stator resistance $R/\Omega$ | 0.36 |
| d- and q-axis inductance $L_{\mathrm{d}}$, $L_{\mathrm{q}}$/mH | 0.201 |
| Permanent magnet flux linkage $\psi_{\mathrm{i}}$/Wb | 0.00655 |
| Number of pole pairs $p$ | 4 |
| Rotational inertia $J$/[Kg·m$^2$] | 0.0000071 |

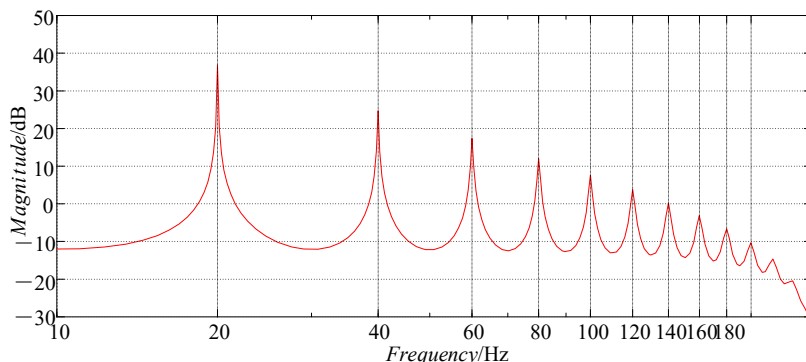

**Figure 3.** The open-loop amplitude–frequency characteristic of the CRC ($N$ = 50).

To effectively suppress the speed ripple at a certain frequency, the open-loop gain in the speed loop controller at that frequency needs to be as large as possible [21]. Figure 3 shows that the CRC can provide a large enough gain at integer multiples of 20 Hz. Therefore, the CRC can perfectly track and suppress a given disturbing signal at specific frequencies.

However, in actual engineering applications, it is difficult to keep the fundamental frequency of the disturbance signal $f_0$ unchanged. When the motor speed changes suddenly, $N$ may be a non-integer value, which means that the fundamental frequency of the signal (speed ripple) that needs to be tracked or suppressed deviates from the fundamental frequency of the CRC, resulting in poor control performance. For example, if the speed reference of the motor changes from 300 [rpm] to 307 [rpm], $f_0$ changes from 20 Hz to 20.47 Hz. With $N$ = *round* $(f_{\mathrm{s}}/f_0)$ = 48, the open-loop amplitude–frequency characteristic of the CRC is shown in Figure 4, where the maximum gain (36.7 dB) appears at a frequency of 20.83 Hz, but the open-loop gain is only 12.5 dB at a frequency of $f_0$ = 20.47 Hz. Although

the frequency deviation is only 0.36 Hz, the gain decreases by 65.9%, which greatly lowers the ability of the CRC to track or suppress periodic signals at specific frequencies.

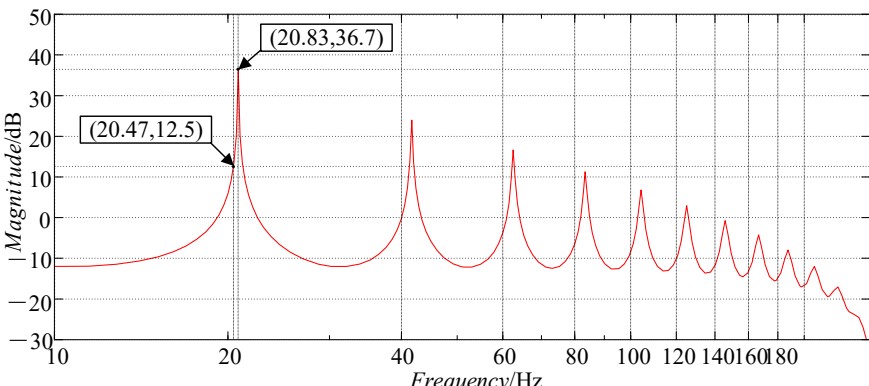

**Figure 4.** The open-loop amplitude–frequency characteristic of the CRC (*N* = 48).

From the above analysis, it can be seen that when the motor speed changes continuously, the CRC has an insufficient disturbance suppression capability under the condition pf *N* not being equal to an integer. The essential reason for this phenomenon is the error introduced when *N* is rounded. The smaller the *N* value (at a higher motor speed), the greater the negative effect.

*2.3. Excessive Speed Overshoot Caused by the CRC*

With the addition of the resonant controller, the gain amplitude around the resonant frequency changes sharply, which easily causes the Nyquist curve to approach the critical point (−1,0). Thus, the sensitivity function of the system is increased, which can aggravate the oscillation of the dynamic regulation process of the system and increase the overshoot [22,23]. The repetitive controller can be considered a combination of a negative proportional gain term, an integral term, and a series of resonant controllers [24]. Therefore, systems with an additional repetitive controller can also lead to a poor dynamic performance and a large overshoot. However, this has rarely been analyzed in the existing literature.

In this paper, the phase compensator adopts the mode of linear phase lead compensation with $C(z) = z^m$, where *m* is the phase lead compensation value, and $z^m$ provides a phase lead angle $\theta = 180° m\omega / \omega_N$ to compensate for the phase lag in the system [25]. $\omega_N$ is the Nyquist frequency. Assuming that *Q*(*z*) is 1, the differential form of the CRC can be deduced from Equation (9), which is given by

$$u(k) = u(k - N) + k_{rc}e(k - N + m),$$  (10)

where *k* = 1,2,3,4 …; *u*(*k*) is the output signal sequence of RC; and *e*(*k*) is the speed deviation sequence.

Assuming that the fundamental period of the disturbance signal to be suppressed is $T_0$ and the sampling period of the controller is $T_s$, it can be seen from Equation (10) that the output signal of the CRC at the $t_k$ moment is not only related to the output signal of the CRC at the $t_k$-$T_0$ moment but also related to the speed deviation at $t_k - T_0 + mT_s$, which is a certain moment near $t_k$-$T_0$. It is assumed that the system is about to reach a steady state at $t_k$, which means that the difference between the speed reference and the actual speed of the system is small. At this point, the output of the CRC should not be too large. Otherwise, it will produce a large overshoot. However, at $t_k - T_0$ and $t_k - T_0 + mT_s$, the system is still in a transient process (such as starting or experiencing a sudden load), that is, the speed deviation at $t_k - T_0 + mT_s$ is large. So, the output signal *u*(*k*) of the CRC at $t_k$ is large, which leads to an abnormal overshoot in the motor speed. Moreover, as the CRC is a kind of periodic controller, the speed overshoot at $t_k$ will continue to affect the controller

output at $t_k + nT_0$ ($n = 1, 2, 3, \ldots$), affecting the actual motor speed at these moments until the speed converges to the speed reference. With a view to reducing the system overshoot, the gain in the repetitive controller should be as small as possible.

A control block diagram of the system with the CRC is presented in Figure 5, where $n_m{}^*(z)$ and $n_m(z)$ are the speed reference and the feedback speed, respectively; $i_q{}^*(z)$ is the q-axis current reference; $P(z)$ is the discrete form of the transfer function $1/(Js + B)$; $PI(z)$ is the transfer function of the speed loop PI controller; $i_{err}(z)$ is the equivalent disturbance caused by the non-ideal factors; $G(z) = k_T \cdot G_c(z) \cdot P(z)$ where $G_c(z)$ is the current loop transfer function; and $k_T$ is the torque coefficient with $k_T = 1.5 \, p\psi_i$.

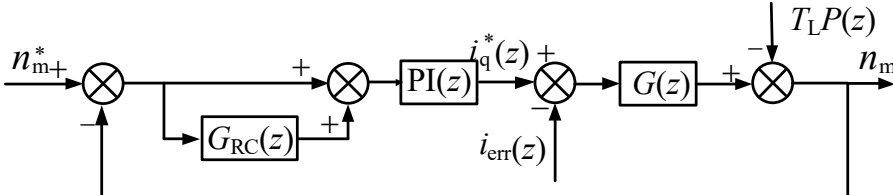

**Figure 5.** Idealized control block diagram of the system with CRC.

Defining $M(z)$ as the transfer function of the system without the repetitive controller, it is given by

$$M(z) = \frac{PI(z)G(z)}{1 + PI(z)G(z)}, \tag{11}$$

Then, the speed deviation $e(z)$ can be given by

$$e(z) = e(z)z^{-N}Q(z)[1 - k_{rc}C(z)M(z)] + \frac{[1 - Q(z)z^{-N}][n_{rn}^*(z) + G(z)i_{err}(z) + T_L P(z)]}{1 + PI(z)G(z)}, \tag{12}$$

When the system is stable, the second term of Equation (12) equals zero. Therefore, in this case, the convergence rate of the speed deviation mainly depends on the first term of Equation (12).

Define $H(z)$ as

$$H(z) = Q(z)[1 - k_{rc}C(z)M(z)], \tag{13}$$

It can be seen from Equations (12) and (13) that the speed deviation can converge to zero after one period of the fundamental wave of the disturbance signal if the design of the gain $k_{rc}$ and the phase compensator $C(z)$ of the repetitive controller satisfies the condition of $k_{rc} \cdot C(z) = M(z)^{-1}$. However, it is impossible to obtain an accurate mathematical model of the transfer function $M(z)$ in practical engineering. In most practical applications, a linear phase compensator is used because it is simple and feasible. If the speed deviation $e(z)$ requires a fast enough convergence rate, $k_{rc}$ should be as large as possible (but no more than $1/|M(z)|$) under the condition that the system is stable.

According to the above analysis, the overshoot and the convergence speed have opposite requirements of $k_{rc}$. Therefore, it is not advisable to solve the problem of speed overshoot simply by reducing the value of $k_{rc}$. A saturation-limiting link is generally added to the output of the repetitive controller to suppress the overshoot problem [16,17,20]. However, this makes a windup phenomenon appear in the system, which ultimately leads to performance degradation [26].

In general, it would be better to formulate such a repetitive controller that in a transient process, when the deviation is large, the gain in the repetitive controller should be appropriately reduced to avoid an overshoot. While in the steady-state convergence stage, that is, when the deviation is small, the gain should be as large as possible (under the condition of system stability) to accelerate the convergence speed.

## 3. The Proposed *fal*-FORC Strategy

### 3.1. Principle of FORC

Many actual systems exhibit fractional order dynamic behavior due to their special physical characteristics. They are fractional order systems. When using a fractional order model to describe an object with fractional order characteristics, it can better reveal the essential characteristics and behavior of the object. Correspondingly, for a fractional order model, a corresponding fractional order controller needs to be designed to improve the control effect [27,28]. In [27], a resonant controller with fractional order calculus is proposed to suppress the periodic current harmonics caused by non-ideal factors in the inverter and current measurement errors. Reference [28] applies fractional order PID to an automatic power generation control system and uses a Crow algorithm to optimize the controller parameters, which improves the dynamic performance and robustness of the system.

A structure diagram of FORC is shown in Figure 6, where $N$ can be calculated by

$$N = \frac{60}{p n_{\mathrm{m}}^* T_{\mathrm{s}}}, \tag{14}$$

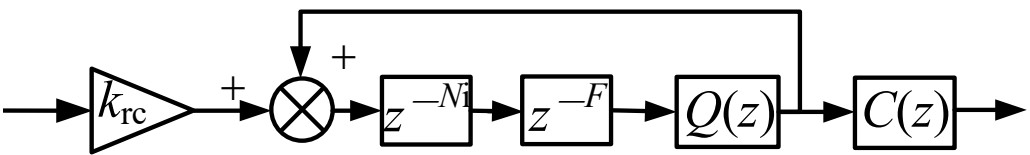

**Figure 6.** A structure diagram of FORC.

When $N$ is a non-integer value, it includes an integer part $N_{\mathrm{i}}$ and a decimal part $F$, that is, $N = N_{\mathrm{i}} + F$. $z^{-F}$ can be fitted as follows:

$$z^{-F} \approx \sum_{k=0}^{n} A_k z^{-k}, \tag{15}$$

where $k = 0, 1, \ldots, n$, and $n$ represents that $z^{-F}$ is fitted using the $n$-order Lagrange interpolation method [29]. The coefficient $A_k$ can be calculated as follows:

$$A_k = \prod_{\substack{i = 0 \\ i \neq k}}^{n} \frac{F - i}{k - i}, \quad k, i = 0, 1, \ldots, n, \tag{16}$$

Then, the Lagrange interpolation remainder is

$$R_{\mathrm{n}} = z^{-F} - \sum_{k=0}^{n} A_k z^{-k} = \frac{\xi^{-F-n} \prod_{i=0}^{n-1} (-F - i)}{(n+1)!} \prod_{i=0}^{n} (F - i), \tag{17}$$

where $\xi \in [T_k, T_{k+1}]$. $T_k$ and $T_{k+1}$ are the $k$th and $k + 1$th sampling instants. It can be seen from Equation (17) that the Lagrange interpolation remainder $R_{\mathrm{n}}$ decreases as $n$ increases. Accordingly, $z^{-F}$ fitted using Equations (15) and (16) is more accurate with a larger $n$.

Figure 7 shows the amplitude–frequency characteristic curve of $z^{-F}$ fitted using the first-order, second-order, and third-order Lagrange interpolation methods. It can be seen from the curve that with an increase in the fitting order, the magnitude of $z^{-F}$ is closer to 1 in a wider frequency range, but the corresponding controller structure is more complex. Therefore, considering the trade-off between accuracy and complexity, this paper chooses to use the second-order Lagrange interpolation method to fit $z^{-F}$.

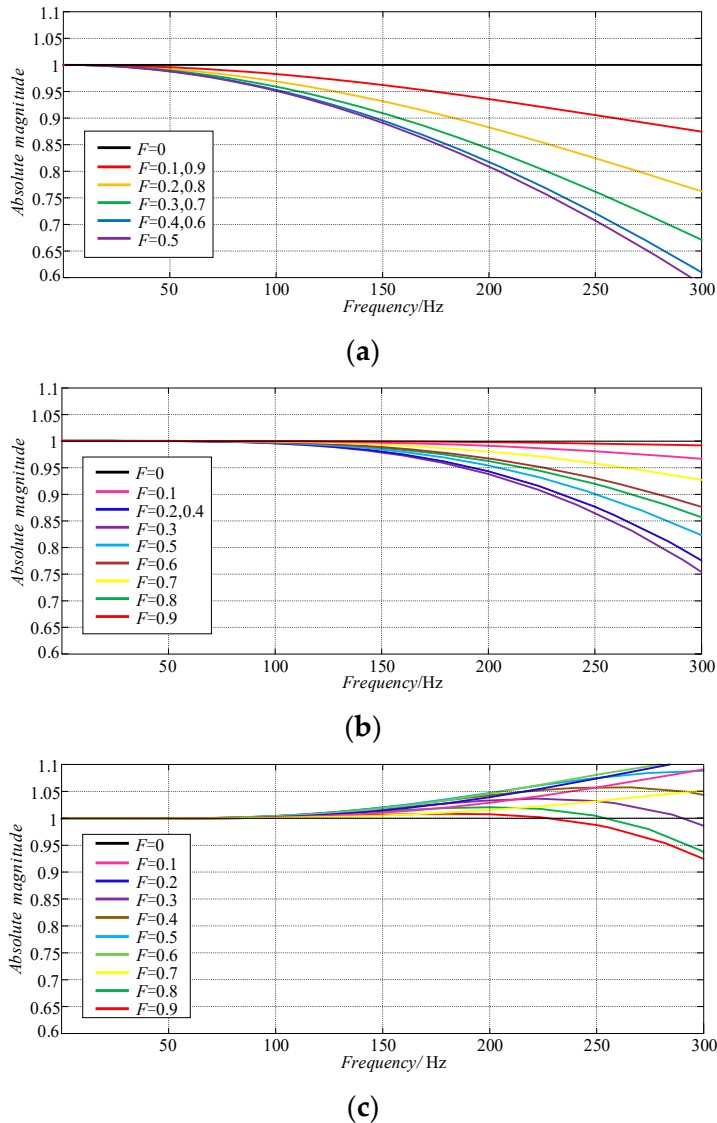

**(a)**

**(b)**

**(c)**

**Figure 7.** The amplitude–frequency characteristics of $z^{-F}$ fitted using Equation (15): (**a**) $n = 1$; (**b**) $n = 2$; (**c**) $n = 3$.

Figure 8 shows the phase–frequency characteristics of $z^{-F}$ fitted using the second-order Lagrange interpolation method. It can be seen that $z^{-F}$ has the characteristics of a linear phase, and its slope gradually approaches the unit delay link $z^{-1}$ when $F$ gradually approaches 1. Such amplitude and phase characteristics are very convenient for the design of other parameters and stability analysis of the repetitive controller.

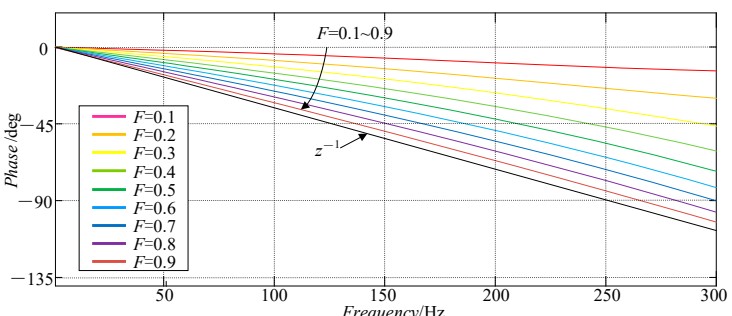

**Figure 8.** The phase–frequency characteristics of $z^{-F}$ fitted using Equation (15) when $n = 2$.

The structure of the FORC used in this paper is shown in Figure 9. Therefore, the transfer function of the FORC is derived as follows:

$$G_{\text{FORC}}(z) = \frac{k_{\text{rc}}C(z)Q(z)z^{-Ni}\sum\limits_{k=0}^{n}A_k z^{-k}}{1 - Q(z)z^{-Ni}\sum\limits_{k=0}^{n}A_k z^{-k}},\tag{18}$$

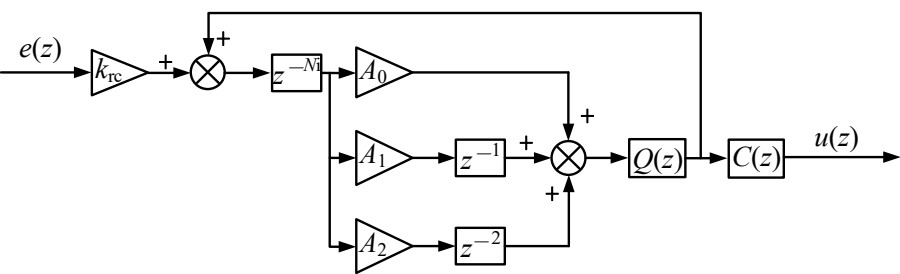

**Figure 9.** The structure of the FORC.

Similarly, if the speed reference of the motor is 307 [rpm] ($f_0$ = 20.47 Hz), it can be deduced that $N = f_s/f_0 = 48.85 = N_i + F$ with $N_i$ = 48 and $F$ = 0.85. According to Equation (10), the open-loop amplitude–frequency characteristic of the FORC is shown in Figure 10. It can be seen from Figure 10 that the maximum gain point for the FORC appears right at the frequency of 20.47 Hz. As analyzed, the FORC can effectively suppress speed ripples when the pulsation period is not an integral multiple of the speed loop sampling period.

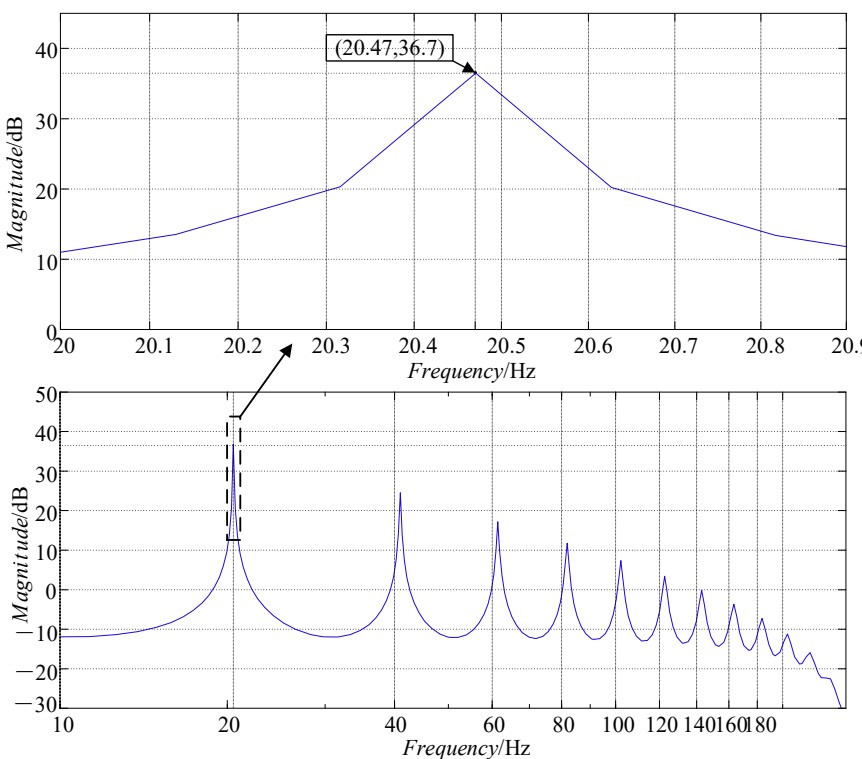

**Figure 10.** The open-loop amplitude–frequency characteristic of the FORC ($N$ = 48.85).

### 3.2. The Proposed fal-FORC and Parameter Determination

To give the repetitive controller the output characteristic analyzed at the end of Section 2, the nonlinear function *fal*($e,\alpha,\delta$) is introduced as the correction factor to adjust

the gain in the FORC dynamically. The structure of the proposed *fal*-FORC is shown in Figure 11, where *fal* stands for the function *fal*(*e*,α,δ), and its expression is

$$fal\,(e,\alpha,\delta) = \begin{cases} \frac{e}{\delta^{1-\alpha}} & |e| \le \delta \\ |e|_{\alpha}\mathrm{sgn}\,(e) & |e| > \delta \end{cases},$$  (19)

where *e* is the input error and δ is the interval length of the linear segment to avoid the high-frequency oscillation caused by large gain when the error is too small [30].

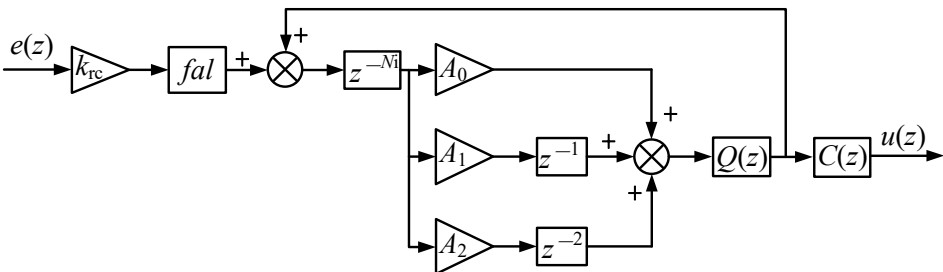

**Figure 11.** The structure of the *fal*-FORC.

The nonlinear function *fal* was first proposed by Han and was well applied to his active disturbance rejection control (ADRC) theory to improve the dynamic performance of a system [31]. In reference [32], the PI parameters are automatically adjusted using the *fal* function to maintain the adaptive ability of the system, improve the dynamic response speed, and enhance the disturbance rejection performance. Reference [33] modifies the sign function in the *fal* function to eliminate the discontinuity of the nonlinear extended state observer and obtains a novel nonlinear observer to measure and estimate the disturbances, uncertainties, and states of the system. However, when the input deviation of the *fal* function is too big, the function gain will be small, which may affect the transient response and anti-interference performance of the controller. Therefore, references [34,35] designed linear/nonlinear switching controllers to avoid this problem. Fortunately, the *fal* function in this article is located in the plug-in repetitive controller, and the system's anti-interference performance still depends on the speed loop PI controller. Therefore, the transient response and anti-interference performance of the system is almost unaffected by the *fal* function. The nonlinearity of *fal*(*e*,α,δ) is determined by α, a constant between 0 and 1. The smaller the value of α, the larger the nonlinearity. With δ = 0.4, α = 0.6. A comparison between the output characteristics of *y* = *fal*(*e*,α,δ) and the linear function *y* = *e* is shown in Figure 12, which shows that when $0 < |e| \le 0.4$, the output of the *fal* function is linearly related to the input error and larger than the input error value. When $0.4 < |e| \le 1$, the relationship between the output and the input of *y* = *fal*(*e*,α,δ) is nonlinear, and the output of *y* = *fal*(*e*,α,δ) is larger than the input error value. When $|e| > 1$, the output of *y* = *fal*(*e*,α,δ) is smaller than the input error value, and the larger the input error, the more obvious the attenuation of the output of the *fal* function. Therefore, the *fal* function is considered to have the characteristics of "big error, small gain, small error, big gain" [35]. This means that when the error is large, a slightly smaller gain is used to avoid overshooting. When the error is small, the gain is increased to avoid a slow approach toward the target value due to too low an error and low gain.

To further study the effect of δ and α on the *fal* function, we define

$$\lambda = \frac{fal\,(e,\alpha,\delta)}{e},$$  (20)

The influence of different δ values on the performance of the *fal* function when α = 0.6 is shown in Figure 13, which reveals that a larger δ can result in a wider linearity range (where λ is a constant greater than 1) of the output of the *fal* function. Moreover, as the value of δ increases, the linear amplifying effect of the *fal* function on small errors becomes

weaker. The purpose of introducing the *fal* function in this paper is to use its nonlinearity to reduce the overshoot of the system, and its nonlinearity is not affected by $\delta$, so the change in $\delta$ has little influence on the ability of the *fal* function to suppress overshoots. In this paper, $\delta = 0.4$ is selected in Section 4.

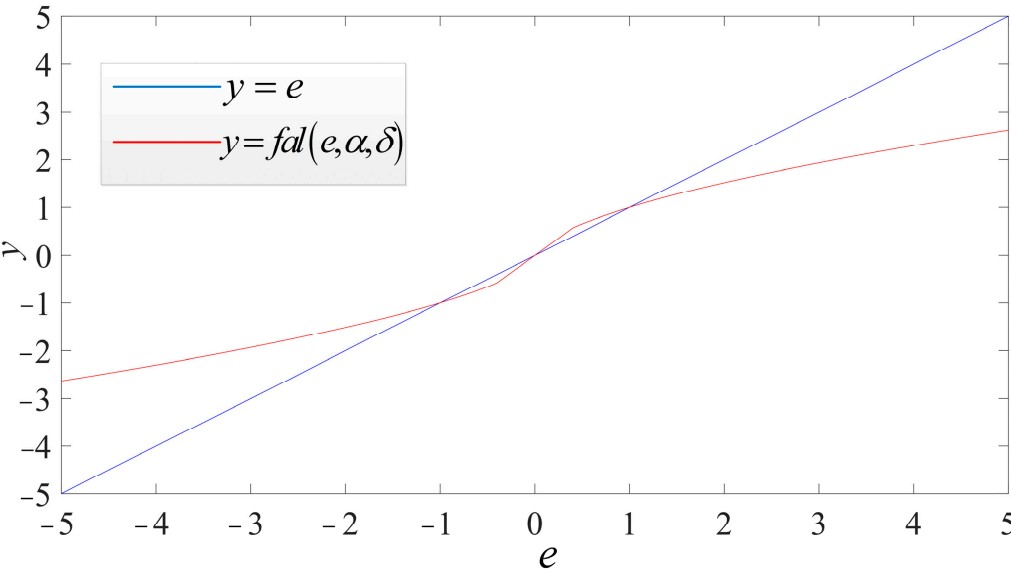

**Figure 12.** The output characteristics of *fal(e,α,δ)* and $y = e$.

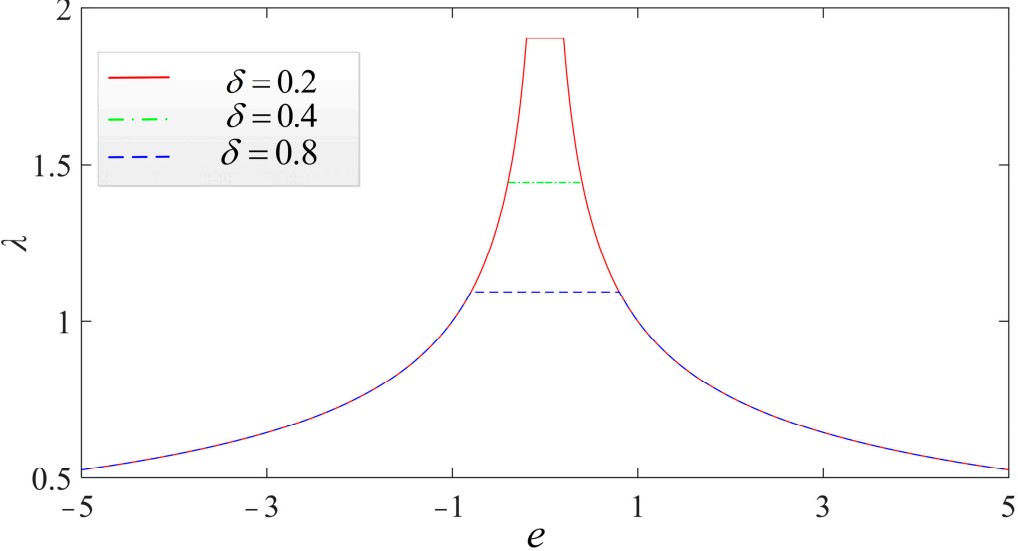

**Figure 13.** Influence of different $\delta$ values on the performance of the *fal* function with $\alpha = 0.6$.

Figure 14 shows the influence of different $\alpha$ values on the performance of the *fal* function when $\delta = 0.4$. It can be seen that the smaller the value of $\alpha$ is, the higher the output nonlinearity of the *fal* function is. In other words, the smaller the value of $\alpha$ is, the stronger the function of *fal* in amplifying small errors, and the stronger its function in reducing large errors. Therefore, too small a value of $\alpha$ will lead to too large a gain in the small error interval, which may cause high-frequency oscillations in the system, while too large a value of $\alpha$ (close to 1) means that the *fal* function loses its ability to reduce the overshoot of the system. In this paper, $\alpha$ is set to 0.6.

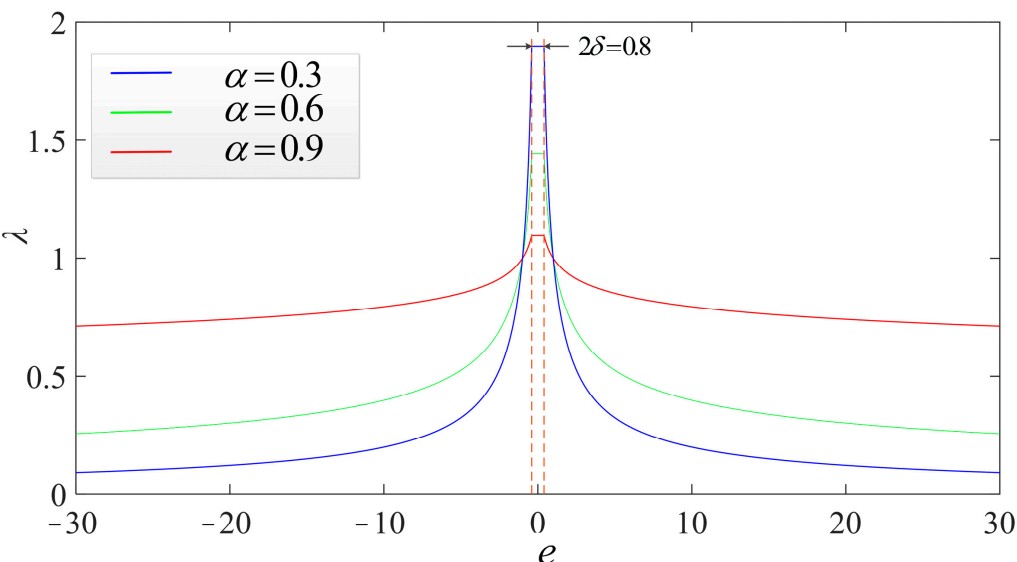

**Figure 14.** $\delta = 0.4$, the effect of different $\alpha$ values on $\lambda$.

The structure of the vector control system of the PMSM drive with the proposed *fal*-FORC strategy in this paper is shown in Figure 15, where the symbols are described in Table 2. This plug-in structure combines the high dynamics of the PI controller with the good steady-state harmonic suppression capabilities of the repetitive controller and minimizes the mutual interference between the two controllers [36]. It is worth noting that when the traditional vector control method is used, the input signal of the speed outer loop PI controller is the deviation value of the speed, and its expression is shown in Equation (21). When the CRC is inserted into the vector control structure, the input signal of the speed outer loop PI controller is shown in Equation (22). When FORC and *fal*-FORC are inserted into the vector control structure, respectively, the input signal expressions of the speed outer loop PI controller are as shown in Equations (23) and (24), respectively.

$$e_1 = n_m^* - n_m, \tag{21}$$

$$e_2 = (1 + \frac{k_{rc}C(z)Q(z)z^{-N}}{1 - Q(z)z^{-N}})(n_m^* - n_m) \tag{22}$$

$$e_3 = (1 + \frac{k_{rc}C(z)Q(z)z^{-Ni}\sum\limits_{k=0}^{n}A_kz^{-k}}{1 - Q(z)z^{-Ni}\sum\limits_{k=0}^{n}A_kz^{-k}})(n_m^* - n_m) \tag{23}$$

$$e_4 = (1 + \frac{\lambda k_{rc}C(z)Q(z)z^{-Ni}\sum\limits_{k=0}^{n}A_kz^{-k}}{1 - Q(z)z^{-Ni}\sum\limits_{k=0}^{n}A_kz^{-k}})(n_m^* - n_m) \tag{24}$$

**Table 2.** Description of symbols in Figure 15.

| Sign | Description |
| --- | --- |
| $i_d^*/i_d$ | Reference/feedback of d-axis current |
| $i_q^*/i_q$ | Reference/feedback of q-axis current |
| $i_{A\_mea}/i_{B\_mea}$ | Feedback of A-phase/B-phase current |
| $i_\alpha/i_\beta$ | Feedback of $\alpha$-axis/$\beta$-axis current |
| $u_d^*/u_q^*$ | Reference of d-axis/q-axis voltage |
| $u_\alpha^*/u_\beta^*$ | Reference of $\alpha$-axis/$\beta$-axis voltage |
| $\theta_e$ | Motor electrical angle |

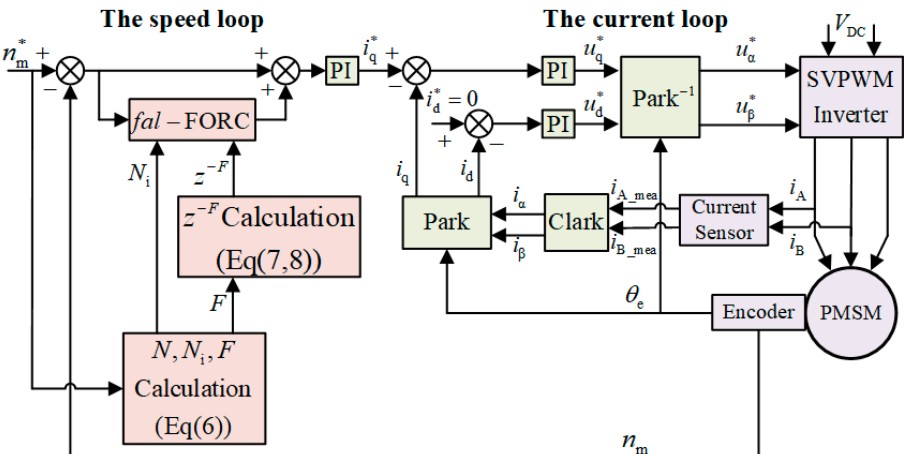

**Figure 15.** Structure of the vector control system of PMSM drive with the proposed *fal*-FORC strategy.

### 3.3. Stability Analysis

As shown in Figures 5 and 11, the speed deviation $e(z)$ can be expressed as follows:

$$
\begin{aligned}
e(z) &= \frac{n_{\mathrm{m}}^*(z) + G(z)i_{\mathrm{err}}(z) + T_{\mathrm{L}}P(z)}{1 + [1 + G_{\mathrm{fal-FORC}}(z)]PI(z)G(z)} \\
&= \frac{1}{1 + PI(z)G(z)} \frac{[z^{N_{\mathrm{i}}+F} - Q(z)][n_{\mathrm{m}}^*(z) + G(z)i_{\mathrm{err}}(z) + T_{\mathrm{L}}P(z)]}{z^{N_{\mathrm{i}}+F} - Q(z)[1 - fal \cdot k_{\mathrm{rc}}C(z)M(z)]}'
\end{aligned}
\tag{25}
$$

The PI parameters are designed according to the method in [11]. The speed loop PI parameters are $k_{\mathrm{sp}} = 0.0368$ and $k_{\mathrm{si}} = 0.92$, and the current-loop PI parameters are $k_{\mathrm{cp}} = 0.6$ and $k_{\mathrm{ci}} = 1080$. This parameter setting can ensure the stability of the traditional PI control system without additional repetitive controllers. All the characteristic roots of $1 + PI(z)G(z) = 0$ are located within the unit circle. Therefore, as long as the characteristic roots of the denominator of the right half of the above equations are located within the unit circle, the stability of the system with a plug-in repetitive controller can be guaranteed. The stability condition can be derived as follows:

$$
|Q(z)[1 - fal \cdot k_{\mathrm{rc}}C(z)M(z)]| = \left| z^{N_{\mathrm{i}}+F} \right| < 1,
\tag{26}
$$

#### 3.3.1. Design of Q(z)

There are normally three forms of $Q(z)$: a constant close to 1 but less than 1, a Butterworth low-pass filter with phase compensation, and a FIR filter with phase compensation. Compared with a constant close to 1, a repetitive control system with a low-pass filter can effectively suppress the ripples within the cut-off frequency of the filter while guaranteeing stability above the cut-off frequency. The advantage of a Butterworth low-pass filter is that there is no steady-state error, and it has sufficient attenuation in the high-frequency band. However, this kind of filter often produces a large phase lag, which has a negative impact on the design of subsequent parameters. Therefore, it is often necessary to design a phase compensator separately for it, which has a complicated structure.

A linear-phase FIR filter is convenient in design and can be precisely compensated for using a non-causal phase lead term to achieve zero phase delay. Its expression can be given as follows:

$$
Q(z) = \sum_{i=0}^{m} a_i z^i + \sum_{i=1}^{m} a_i z^{-i},
\tag{27}
$$

where $a_0 + 2\sum_{i=1}^{m} a_i = 1$. The larger the value of $a_0$ is, the lower the cutoff frequency of the filter is. The filter designed in this article is

$$
Q(z) = 0.45z^{-1} + 0.1 + 0.45z,
\tag{28}
$$

### 3.3.2. Design of C(z)

Since $Q(z)$ is close to 1 at low frequencies, the stability condition in Equation (26) can be simplified as follows:

$$|[1 - fal \cdot k_{rc}C(z)M(z)]| < 1, \tag{29}$$

where $M(z)$ can be expressed as $M(e^{j\omega}) = N_M(e^{j\omega})\exp(j\theta_M(e^{j\omega}))$ and $C(z)$ can be expressed as $C(e^{j\omega}) = N_C(e^{j\omega})\exp(j\theta_C(e^{j\omega}))$ [25]. $N_M(e^{j\omega})$ and $N_C(e^{j\omega})$ are magnitude characteristics. $\theta_M(e^{j\omega})$ and $\theta_C(e^{j\omega})$ are phase characteristics. Substitute $M(e^{j\omega})$ and $C(e^{j\omega})$ into Equation (29), and the following expression is obtained:

$$\left|\left(1 - fal \cdot k_{rc}N_M(e^{j\omega})N_C(e^{j\omega})\exp(j\theta_M(e^{j\omega}) + j\theta_C(e^{j\omega}))\right)\right| < 1, \tag{30}$$

where $k_{rc}$, $N_M(e^{j\omega})$, and $N_C(e^{j\omega})$ are all positive values. Hence, the plug-in repetitive control system will be stable when the following two conditions are satisfied.

$$0 < fal \cdot k_{rc} < \frac{2\min(\cos(\theta_M(e^{j\omega}) + \theta_C(e^{j\omega})))}{\max(N_M(e^{j\omega})N_C(e^{j\omega}))}, \tag{31}$$

$$\left|\theta_M(e^{j\omega}) + \theta_C(e^{j\omega})\right| < 90°, \tag{32}$$

Thus, it can be seen that the phase compensation is significant for the stability of the plug-in repetitive control system. In this paper, the phase compensator adopts the mode of linear phase lead compensation with $C(z) = z^m$, where $m$ is the phase lead compensation value and $z^m$ provides a phase lead angle to compensate for the phase lag in the system [25].

Figure 16 is a Bode diagram of $z^m M(z)$ under different $m$ values. It can be seen that when $m$ is selected between 2 and 6, the stability requirement of Equation (32) can be satisfied. In order to enable $z^m$ to compensate for the phase angle to about 0 degrees in the low-frequency band, $m = 5$ is selected.

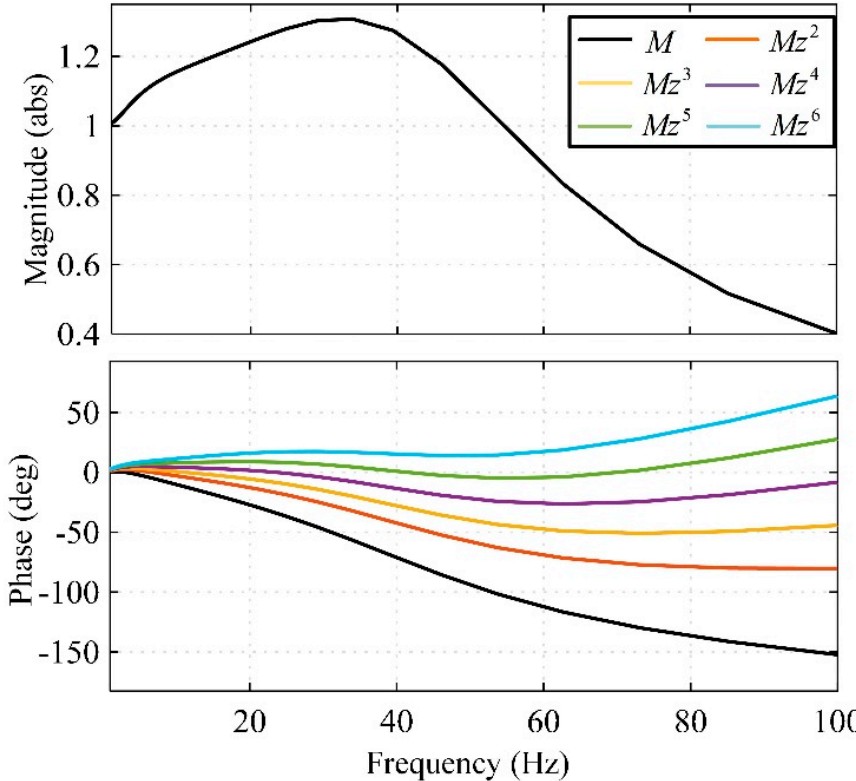

**Figure 16.** Bode diagram of $z^m M(z)$ with different $m$.

### 3.3.3. Design of $k_{rc}$

The RC gain $k_{rc}$ should satisfy the condition in Equation (31). According to Figure 16, it can be seen that when $m = 5$, the following expression can be obtained:

$$\max(N_M(e^{j\omega})) = 1.31, \tag{33}$$

$$\min(\cos(\theta_M(e^{j\omega}) + \theta_C(e^{j\omega}))) = \cos(27.4°) = 0.888, \tag{34}$$

Substituting Equations (33) and (34) into Equation (31), the following expression can be obtained:

$$0 < fal \cdot k_{rc} < 1.355, \tag{35}$$

As shown in Figure 13, when $\alpha = 0.6$ and $\delta = 0.4$, the maximum value of the *fal* function is 1.4. Thus, when the stability condition is satisfied, the value range of $k_{rc}$ is

$$0 < k_{rc} < 0.968 \tag{36}$$

### 3.4. Simulation Results

To verify the correctness of the above analysis, the Simulink software is used to build the motor control simulation model of the four methods PI, CRC, FORC, and *fal*-FORC mentioned above, and the effectiveness of the proposed method is verified using a simulation comparison. The parameters of the PMSMs are listed in Table 1. The artificial current measurement errors are added to the motor drive system with $k_a = 1.1$, $k_b = 0.9$, $\Delta I_{A-offset} = 0.2$ A, and $\Delta I_{B\_offset} = 0.05$ A. The sampling frequency of the speed loop is 1 kHz. The current loop PI parameters are adjusted by the current loop bandwidth $\omega_{cu}$: $k_{cp} = \omega_{cu}L_{dq}$, $k_{ci} = \omega_{cu}R$. In this paper, the current loop bandwidth $\omega_{cu}$ is selected as 2100 [rad/s].

The simulation results are as follows: Figure 17 shows the curves of the speed response with the CRC, with FORC, and without RC under changing speed references, which are 150 [rpm], 203 [rpm], 255 [rpm], and 295 [rpm], respectively. Figure 18 shows the corresponding q-axis current curves. Figure 19 shows the speed responses with *fal*-FORC and FORC at different speed references.

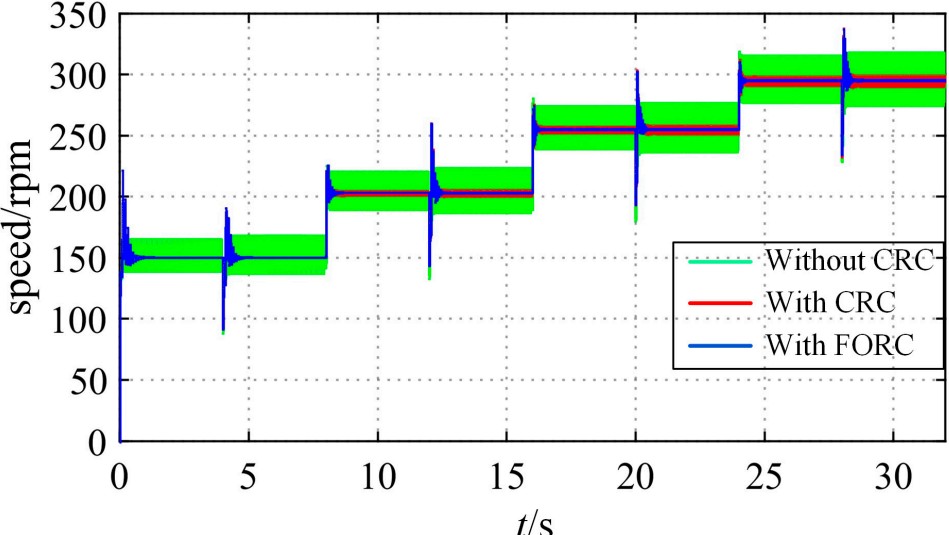

**Figure 17.** Curves of the speed response with CRC, with FORC, and without RC under changing speed references (with sudden loading at t = 4 s, 12 s, 20 s, 28 s).

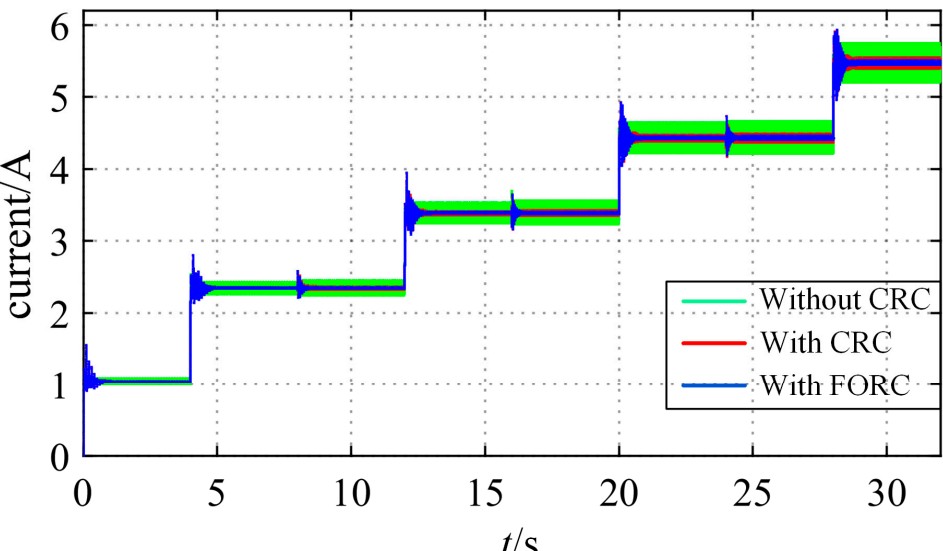

**Figure 18.** Curves in the q-axis current response with CRC, FORC, and without RC under changing speed references (with sudden loading at t = 4 s, 12 s, 20 s, 28 s).

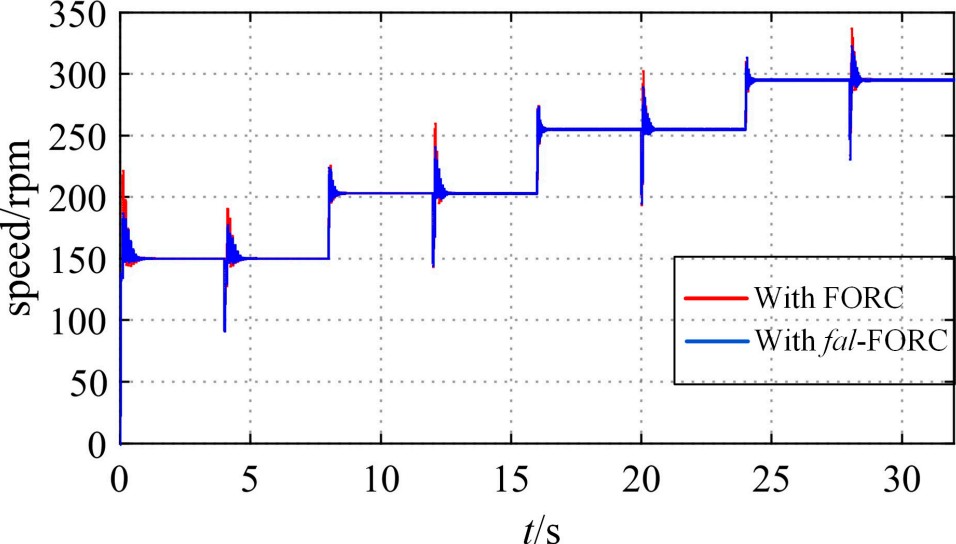

**Figure 19.** Speed responses with *fal*-FORC and FORC at different speed references (with sudden loading at *t* = 4 s, 12 s, 20 s, 28 s).

As can be seen from Figure 17, when the speed reference is 150 [rpm], $N = 100$, in this case, the CRC can well reduce the motor's steady-state speed ripples, and it has the same effect as the FORC. However, in the other three cases, the values of $N$ are non-integers, and it is apparent that FORC presents better results than the CRC in suppressing the speed ripples. Similarly, as shown in Figure 18, when $N$ is not an integer, the q-axis current ripple suppression effect of FORC is better than that of the CRC.

Figure 19 shows a comparison of the speed response when applying the proposed *fal*-FORC strategy and the FORC strategy. It can be seen that when the speed reference is 150 [rpm], the overshoots in the motor speed can be effectively reduced using *fal*-FORC. In addition, the simulation results indicate that the speed drop in the motor when loading is almost the same when using these four methods. This is because the proposed method aims to reduce the speed ripples, and the motor's anti-interference performance still relies on the PI controller.

## 4. Experimental Results

The experiments are implemented for two purposes. One is to verify that compared with the CRC, the FORC has a better suppression ability for speed ripples for which the period is not an integral multiple of the speed loop sampling period. The other is to verify that the vector control system using the proposed *fal*-FORC has a smaller overshoot than that using the FORC only during the stages of starting and sudden loading. The experimental platform is shown in Figure 20, which adopts the form of a pair of PMSMs hauling each other. The parameters of the PMSMs are listed in Table 1. The model of the motor control system is established using Simulink and then transformed into C code using the code automatic generation tool. The generated C code is downloaded to the F28379D chip of the TI company. After adding *fal*-FORC, the algorithm execution time of the speed outer loop is 2335 clock cycles longer than the traditional PI algorithm, requiring approximately 11.68 μs of execution time. The sampling period of the speed outer loop is 1 ms, and the sampling period of the current inner loop is 0.1 ms, which means that there is enough time to implement the improved method proposed in this article. The motor position and speed information is provided using a 1000-line incremental encoder mounted onto the non-drive end of the PMSM. The experimental results are collected on the monitoring computer using the developed communication program.

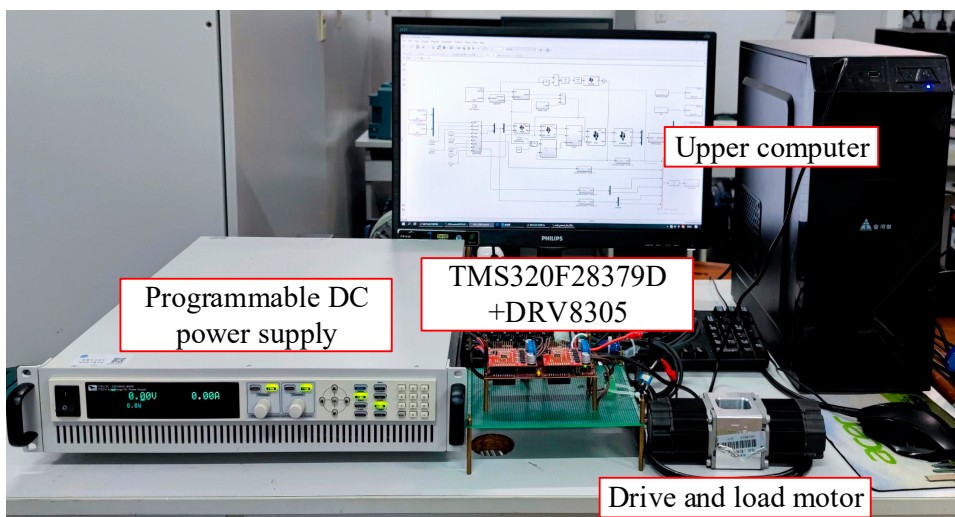

**Figure 20.** The experimental platform.

### 4.1. Verification of the Effectiveness of the FORC Strategy

The experimental results under different values of *m* at a speed of 300 [rpm] are shown in Figure 21. The artificial current measurement errors are added to the motor drive system with $k_a = 1.1$, $k_b = 0.9$, $\Delta I_{A-\text{offset}} = 0.2$ A, and $\Delta I_{B\_\text{offset}} = 0.05$ A. $k_a$ and $\Delta I_{A\_\text{offset}}$ and $k_b$ and $\Delta I_{B\_\text{offset}}$ are the A- and B-phase current scaling and offset errors, respectively. It can be seen from Figure 21 that the system has a relatively fast response when $m = 5$. Thus, $C(z) = z^5$ is adopted in this paper.

Figure 22 shows the experimental results under different values of $k_{\text{rc}}$ at 300 [rpm]. It can be seen that the convergence speed of the system response becomes fast with an increase in $k_{\text{rc}}$. But the consequent overlarge speed overshoot is another consideration. Hence, the value of $k_{\text{rc}}$ is finally selected as 0.6 for a compromise.

In fact, the method proposed in this article reduces the speed ripple by suppressing the motor q-axis current ripple (electromagnetic torque ripple). To verify this, Figure 23 shows the curves of the speed and q-axis current response with the CRC, with FORC, and without RC under changing speed references, which are 150 [rpm], 203 [rpm], 255 [rpm], and 295 [rpm], respectively. The sampling frequency of the speed loop is 1 kHz, and the corresponding values of *N* in the four cases are 100, 73.89, 58.82, and 50.84, respectively. In Figures 23 and 28, the motor is started with loading (the current is about 1 A at this

time, corresponding to 15% of the motor's rated torque). The first loading is 21% of the rated torque, and the current changes by about 1.5 A. The remaining three cases are all loaded at 15%. The rated torque and current are each increased by 1 A, and the final load is approximately 81% of the rated torque. As can be seen from Figure 23, when the speed reference is 150 [rpm], $N = 100$, in this case, the CRC can well reduce the motor steady-state speed and q-axis current ripples, and it has the same effect as the FORC. However, in the other three cases, the values of $N$ are non-integers, and it is apparent that FORC presents better results than the CRC in suppressing the speed and q-axis current ripples.

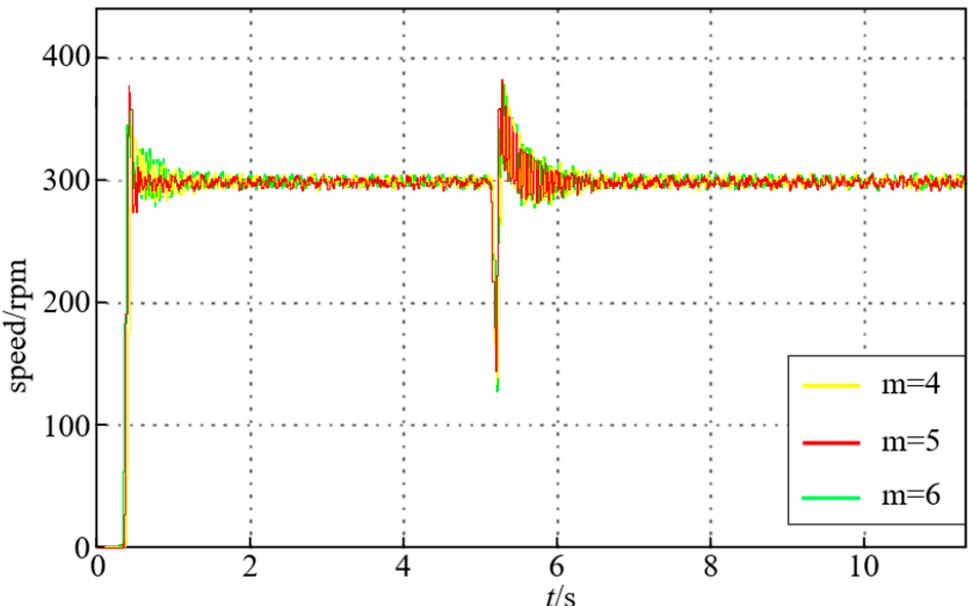

**Figure 21.** Experimental results under different values of $m$ at 300 [rpm].

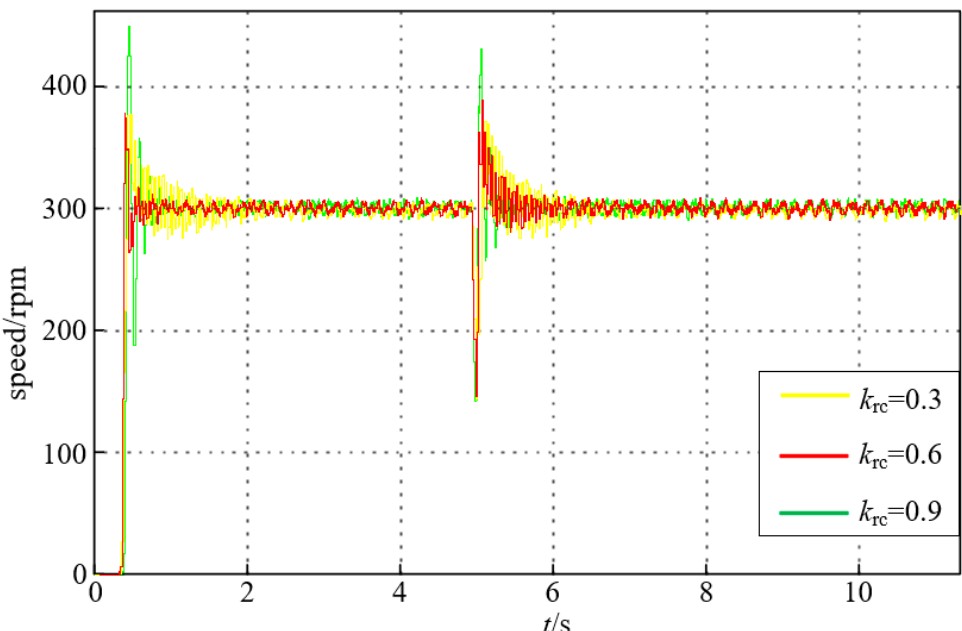

**Figure 22.** Experimental results under different values of $k_{rc}$ at 300 [rpm].

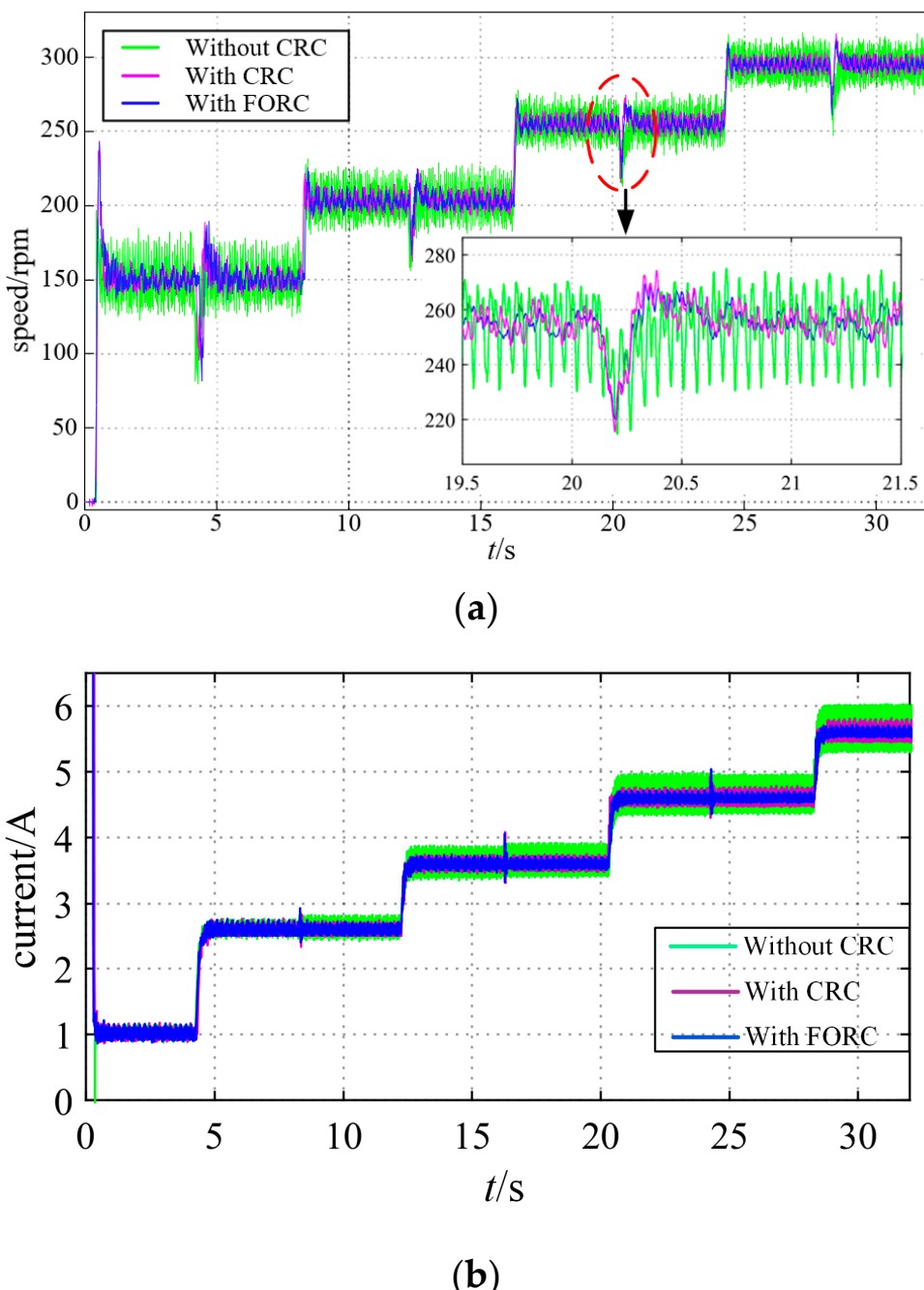

**Figure 23.** Curves of the response with CRC, with FORC, and without RC under changing speed references (with sudden loading at *t* = 4 s, 12 s, 20 s, 28 s). (**a**) Speed. (**b**) Q-axis current.

The FFT analysis results on the steady-state speed with four different speed references are shown in Figure 24, Figure 25, Figure 26, and Figure 27, respectively. It can be seen that when the speed reference is 150 [rpm], the CRC and FORC can both reduce the first-order and second-order pulsation components in the steady-state speed from 11.8% and 5.8% to both less than 0.5%. However, as shown in Figure 25, when the speed reference increases to 203 [rpm], *N* is no longer an integer, and there are still small amounts of the first-order and second-order pulsation components (1.12% and 0.74%) remaining in the steady-state speed of the motor when applying the CRC strategy. In contrast, with the strategy of FORC, the first-order and second-order pulsation components are only 0.17% and 0.16%. The FFT analysis results in Figures 26 and 27 can also prove that FORC performs better than

the CRC when the speed pulsation period is not an integer multiple of the speed loop sampling period.

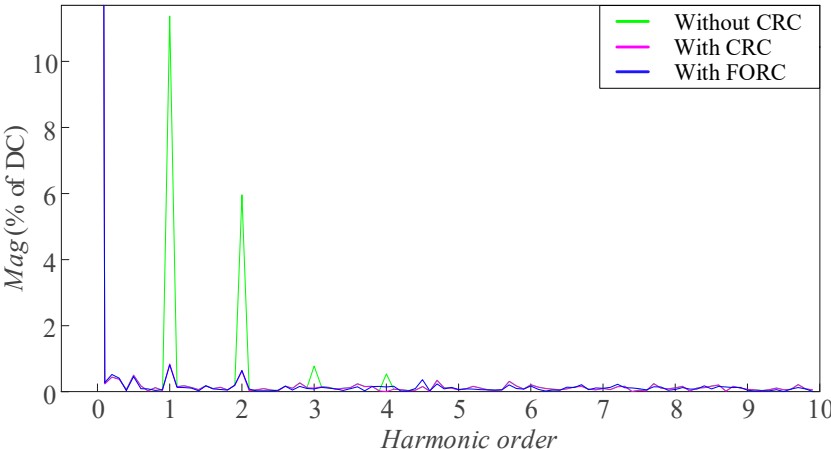

**Figure 24.** FFT analysis of the steady-state speed when the speed reference is 150 [rpm].

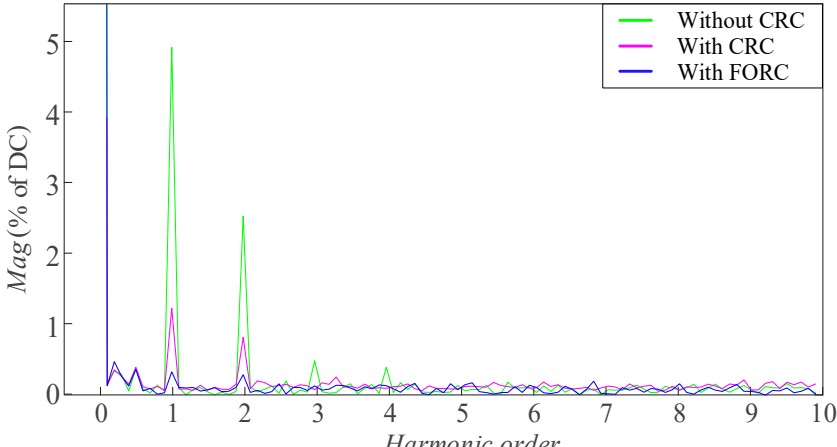

**Figure 25.** FFT analysis of the steady-state speed when the speed reference is 203 [rpm].

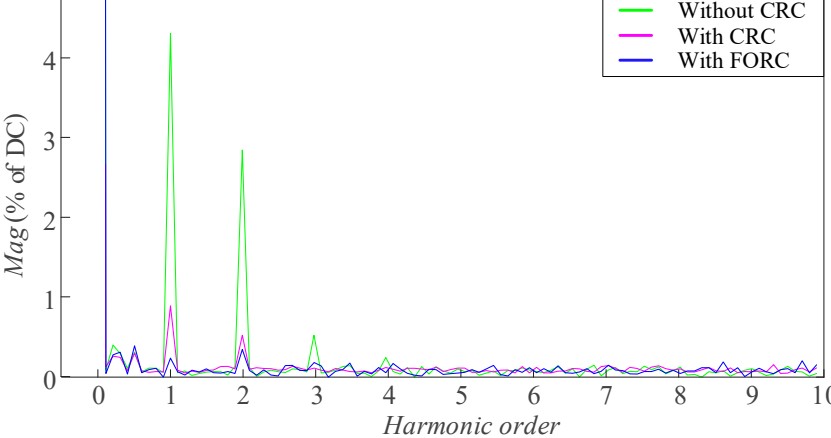

**Figure 26.** FFT analysis of the steady-state speed when the speed reference is 255 [rpm].

To further prove the performance of the FORC at higher speeds when $N$ is a non-integer, the speed references are continuously increased from 0 to 300 [rpm], 367 [rpm], 430 [rpm], and 488 [rpm], and the corresponding values of $N$ are 50, 40.87, 34.88, and 30.74, respectively. It can be seen from Figure 28 that when the speed reference is 300 [rpm], the

FORC and CRC have the same effect on the speed and q-axis current ripple suppression. In contrast, when the speed increases and *N* is no longer an integer, FORC's ability to suppress speed and q-axis current ripples is significantly stronger. By comparing Figures 23 and 28, it can be seen that the higher the motor speed, the more significant the ripple suppression effect of FORC when compared with the CRC. This is because the higher the motor speed, the smaller the value of *N*. *N* is rounded to the nearest integer for the strategy with the CRC, and the smaller the value of *N*, the larger the relative deviation caused by rounding. In this case, the suppression effect of the CRC is in sharp contrast to that of FORC.

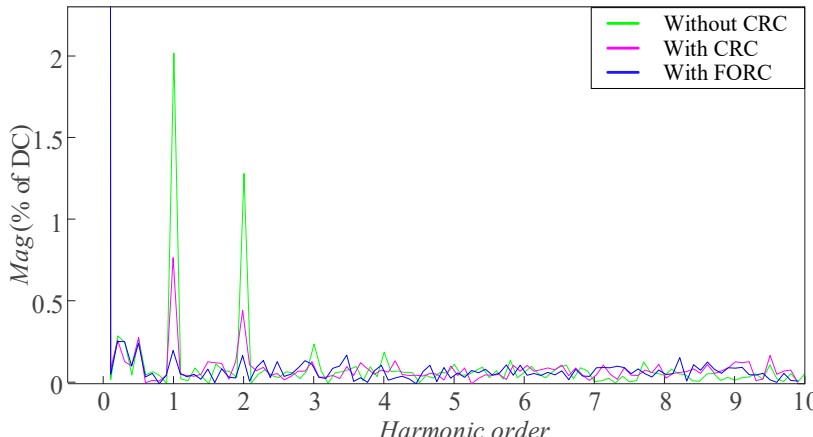

**Figure 27.** FFT analysis of the steady-state speed when the speed reference is 295 [rpm].

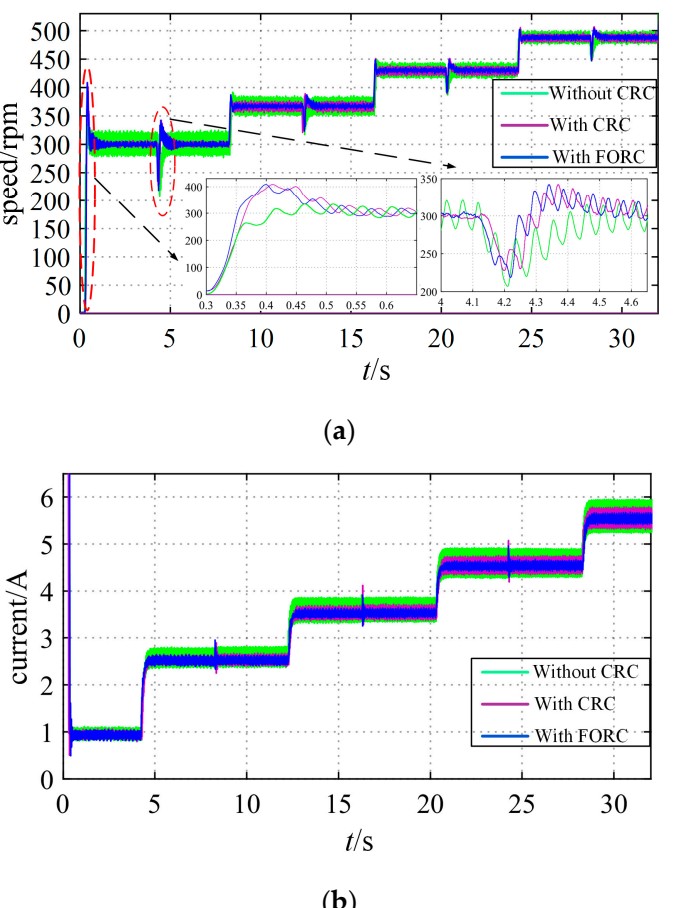

(**a**)

(**b**)

**Figure 28.** Curves of the response with CRC, with FORC, and without RC under higher speed references (with sudden loading at *t* = 4 s, 12 s, 20 s, 28 s). (**a**) Speed. (**b**) Q-axis current.

Due to the filtering effect of the motor shaft inertia, high-frequency harmonics are difficult to reflect in the motor speed. To illustrate this issue, a set of higher-speed experiments are conducted. The reference speeds are set to 900 [rpm] and 1200 [rpm], respectively, and then the load of 21% rated torque is increased at 2.3 s and 6.3 s, respectively. The experimental results are shown in Figure 29. It can be seen that the motor speed ripple is very small. At this time, the current measurement error is mainly reflected in the q-axis current ripple. When the rotational speed is 1200 [rpm], the proposed method can effectively suppress the q-axis current ripple, but the rotational speed pulsation at this time is almost the same as before the improvement.

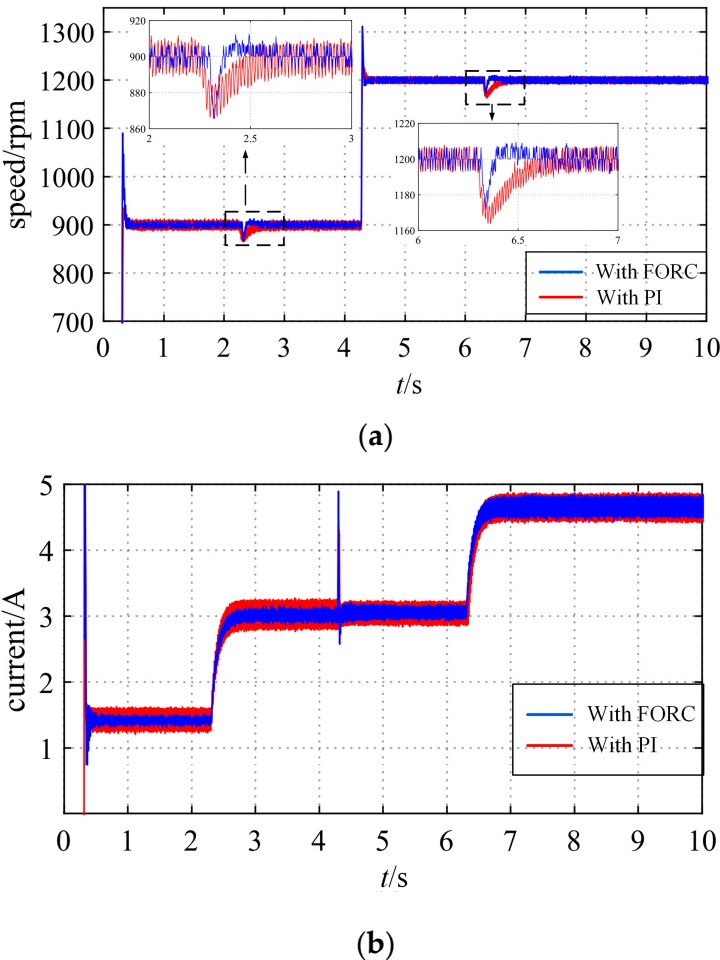

**(a)**

**(b)**

**Figure 29.** Responses with PI and FORC at two higher speed references (with sudden loading at $t = 2.3$ s, 6.3 s). (**a**) Speed. (**b**) Q-axis current.

### 4.2. Effectiveness of the fal-FORC Strategy

Figures 30 and 31 show a comparison of the speed response when applying the proposed *fal*-FORC strategy and the FORC strategy. It can be seen that when the speed reference is 150 [rpm], the overshoots in the motor speed with FORC during startup and loading are 95 [rpm] (63.3%) and 46 [rpm] (30.7%), respectively, which are obviously unacceptable. In contrast, the speed overshoots of the motor with the *fal*-FORC strategy are 29 [rpm] (19.3%) and 26 [rpm] (17.3%), respectively. Obviously, the speed overshoots can be effectively reduced using *fal*-FORC. Similarly, it can be seen from Figure 31 that when the speed reference is 300 [rpm], the speed overshoots of the motor with the *fal*-FORC strategy during motor starting and loading are 65 [rpm] (21.7%) and 41 [rpm] (13.7%) respectively, which are smaller than the overshoots of 119 [rpm] (39.7%) and 46 [rpm] (15.3%) when using FORC. Therefore, the proposed *fal*-FORC strategy is an effective control strategy for reducing overlarge speed overshoots. In addition, the experimental results

in Section 4 demonstrate that the anti-interference performance of the motors using the four methods is almost the same, and the speed drop during loading is basically the same. The anti-interference performance of the motor still relies on the PI controller, which is consistent with the simulation analysis.

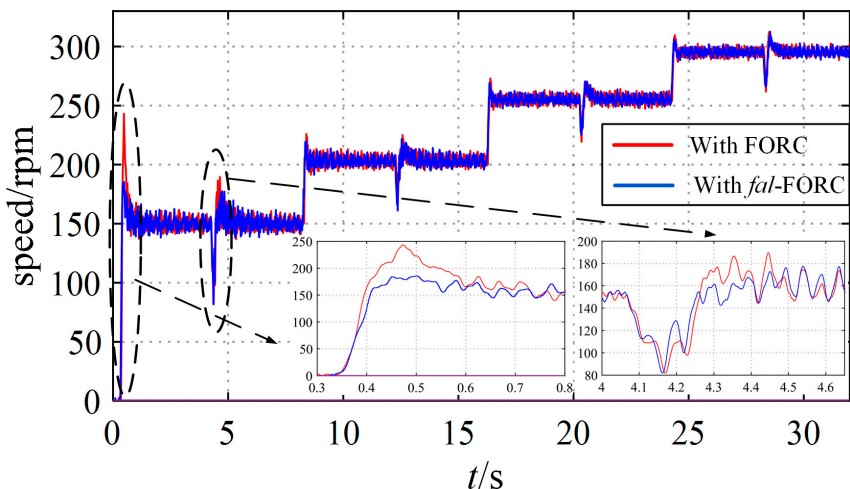

**Figure 30.** Speed responses with *fal*-FORC and FORC at different speed references (with sudden loading at *t* = 4 s, 12 s, 20 s, 28 s).

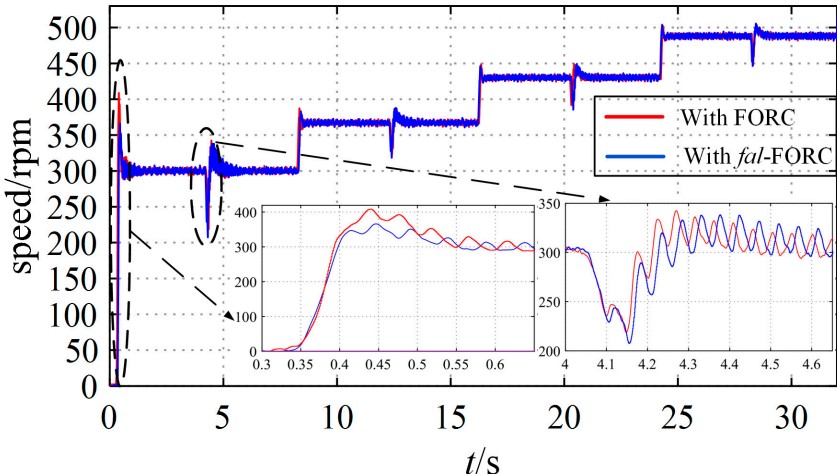

**Figure 31.** Speed responses with *fal*-FORC and FORC at four higher speed references (with sudden loading at *t* = 4 s, 12 s, 20 s, 28 s).

To analyze the difference in the characteristics between the simulation and experimental results, Table 3 records the FFT analysis results and speed overshoot values of the simulation and experimental results. The FFT analysis data are all simulation or experimental results from when the motor speed is 255 [rpm]. The results show that after adopting the compensation method, the speed harmonics in both the simulation and experiment are significantly reduced, and the q-axis current harmonics in the simulation are also significantly reduced. In the experiment, the q-axis current harmonics can be reduced by about 80%. This is because in the actual motor system, in addition to the current measurement error, which is the main source of the first and second harmonics, there will also be a pulsating component in the load torque. The proposed method will generate the opposite current pulsating component to offset its influence and suppress the speed ripple. But the current ripple still exists. The simulation results on the speed overshoot are basically consistent with the experimental results. The *fal* function can effectively reduce the speed overshoot caused by repetitive controllers.

**Table 3.** Comparison results of Figures 17–19, Figures 23 and 30.

|  | Method | Harmonic Content | Speed | q-Axis Current | Speed Overshoot [rpm] |
|---|---|---|---|---|---|
| Simulation | PI | 1st | 4.89% | 3.26% | 15 |
|  |  | 2nd | 3.10% | 4.12% |  |
|  | CRC | 1st | 0.51% | 0.34% | 70 |
|  |  | 2nd | 0.71% | 0.95% |  |
|  | FORC | 1st | 0.03% | 0.02% | 71 |
|  |  | 2nd | 0.09% | 0.13% |  |
|  | *fal*-FORC | 1st | 0.03% | 0.03% | 35 |
|  |  | 2nd | 0.09% | 0.12% |  |
| Experiment | PI | 1st | 4.34% | 4.77% | 25 |
|  |  | 2nd | 2.92% | 3.21% |  |
|  | CRC | 1st | 0.87% | 1.73% | 90 |
|  |  | 2nd | 0.48% | 0.86% |  |
|  | FORC | 1st | 0.20% | 1.12% | 95 |
|  |  | 2nd | 0.33% | 0.50% |  |
|  | *fal*-FORC | 1st | 0.19% | 1.08% | 29 |
|  |  | 2nd | 0.31% | 0.46% |  |

## 5. Conclusions

The *fal*-FORC strategy proposed in this paper can well solve the two problems generated by the CRC strategy in the suppression of the periodic speed fluctuations caused by non-ideal factors in a PMSM drive system. The contributions of this paper are presented as follows:

(1) A theoretical analysis of the two problems, the unsatisfactory ripple suppression of the CRC under variable speeds and the overlarge speed overshoot caused by the CRC, are elaborated on before the design of *fal*-FORC.

(2) A fractional order delay link is introduced to solve the first problem that the CRC strategy has a worse performance in suppressing the periodic speed ripples at frequencies of non-integer multiples of the fundamental frequency. The Lagrange interpolation method is used to fit the fractional delay term. The simulation and experimental results can verify the effectiveness of the FORC strategy.

(3) The nonlinear function *fal*($e,\alpha,\delta$) is designed before FORC, and the basis for the parameter selection of the controller is given to ensure the stability of the system. This method can dynamically adjust the gain in the repetitive controller, effectively reduce the speed overshoot caused by excessive open-loop gain, thereby improving the transient process of motor starting and loading.

The repetitive controller attached to the outer speed loop can effectively suppress the motor speed ripple, but it cannot suppress the periodic disturbance in the d-axis and has a limited improvement effect on the phase current distortion. How to suppress the periodic disturbance introduced by the current measurement error in the d-axis feedback channel is one of our future research directions.

**Author Contributions:** Conceptualization, H.G. and F.Z.; methodology, H.G., F.Z. and Q.Z.; software, H.G. and F.Z.; validation, H.G., F.Z. and Q.Z.; formal analysis, H.G. and F.Z.; investigation, H.G., F.Z. and Y.L.; resources, H.G.; data curation, H.G. and F.Z.; writing—original draft preparation, H.G., F.Z. and J.X.; writing—review and editing, H.G., F.Z. and T.X.; visualization, H.G., F.Z. and T.X.; supervision, H.G., F.Z. and Y.L.; project administration, H.G., F.Z. and Y.L.; funding acquisition, H.G., F.Z., Y.L. and Q.Z.; All authors have read and agreed to the published version of the manuscript.

**Funding:** This research is funded by: (a) The National Key R&D Program of PR China (Grant No. 2023YFB4301704); (b) National Natural Science Foundation of China (Grant No. 52201354, 51979021); (c) Fundamental Research Funds for the Central Universities (Grant No. 3132023621).

**Institutional Review Board Statement:** Not applicable.

**Informed Consent Statement:** Not applicable.

**Data Availability Statement:** Data is contained within the article.

**Conflicts of Interest:** The authors declare that they have no known competing financial interests or personal relationships that could have appeared to influence the work reported in this paper.

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
