# Peer review of "A Combined Fractional Order Repetitive Controller and Dynamic Gain Regulator for Speed Ripple Suppression in PMSM Drives"

_actuators, doi:10.3390/act13020073_

Round 1
Reviewer 1 Report
Comments and Suggestions for Authors
The authors well summarized the problems of previous studies and their own research. The authors then propose an improved fal-FORC strategy and the study's differentiation is relevant.
1. It is necessary to consider the differences in characteristics between analysis results and experiments.
Please write a separate table to compare the errors between the authors' simulation results and the experimental results.
And I hope you will provide supplementary explanation on this.
2. For FORC, the harmonic components in Figure 20 to Figure 23 have been improved. What are the main reasons? Has the waveform of the input current changed? I think it would be good if you could provide further explanation.
Author Response
Thank you very much for your valuable comments, which make our manuscript more perfect. Please see the attachment for detailed modifications.

Reviewer 2 Report
Comments and Suggestions for Authors
This article uses a fractional-order repetitive controller and dynamic gain regulator to suppress the speed ripple and reduce the transient overshoot. The opinions are as follows:
1. A suitable controller should be able to have a good noise suppression effect for a wide range of reference inputs and should not only have this effect for a specific frequency.
2. In line 94, "The current measurement error and other non-ideal factors can cause the 1st-, 2nd-, 3rd-, 4th-...nth-order speed fluctuating components, which can be well suppressed by the strategy with a plug-in repetitive controller." It is unclear here, what are 1st-, 2nd-, 3rd-, 4th-…nth-order speed fluctuating components? How do we define current measurement error? How do we know there will be an error in the measured current?
3. The variables in Eq. (2) should be lowercase because they are discrete-time time-domain signals.
4. The citation no. of Lines 113, 141, and 147 should be capitalized.
5. Please re-examine and correct the variable subscripts throughout the article.
6. Lines 275-276, it said that: Therefore, the fal-function is considered to have the characteristics of "A big error with a small gain, while a small error with a large gain." Is this just like P control?
7. Lines 285-286, "The purpose of introducing the fal-function in this paper is to use its nonlinearity to reduce the overshoot of the system." When overshoot is suppressed, will there be a counter-effect?
8. Theta_c in Figure 14 has no description.
9. Line 386, what is offset error?
10. Line 386, Figure 16 should be Figure 17.
11. What are the encoder specifications of the motor system used in the experiment? How do you get the motor speed in your experiment?
12. Please provide the value of the sudden load.
13. Please provide the input signal of the speed PI controller in Figure 14, including without CRC, with CRC, with FORC, and with fal-FORC, respectively.
14. Please recheck the arrow in Figure 14.
15. Including Figures 19 and 24-26, except that the overshoot and speed ripple are relatively small, please also compare the transient response and anti-interference ability.
16. Mentioned in conclusions, "The repetitive controller attached to the outer speed loop can effectively suppress the motor speed pulsation, but it cannot suppress the periodic disturbance of the d-axis, and has a limited improvement effect on the phase current distortion. How to suppress the periodic disturbance introduced by the current measurement error in the d-axis feedback channel is one of our future research directions." What is "it cannot suppress the periodic disturbance of the d-axis"? It needs to be clearer. What is the current measurement error in the d-axis feedback channel? What is the impact? How do we reduce measurement error in the d-axis feedback channel? Why is there a measurement error in the d-axis feedback channel?
17. Please discuss why the speed ripple without CRC (i.e., only the PI controller) is so large in Figure 19.
18. Why choose 300rpm as the primary operating speed?
19. r.min-1=rpm, microseconds=μs, millisecond=ms.
20. what is the sampling period of the current control loop in the experiment system?
Comments on the Quality of English LanguageAt an average level, it could be better.
Author Response

(The authors gave the same response as above.)

Reviewer 3 Report
Comments and Suggestions for Authors
The manuscript entitled: "Combined Fractional-Order Repetitive Controller and Dynamic gain Regulator for Speed Ripple Suppression of PMSM Drive" concern the problem of control to control the permanent magnet synchronous motors (PMSM). The manuscript can be interested for the readers, due to real problems in control of PMSM. To solve the problem with control the PMSM the authors proposed fractional-order repetitive control (FORC).
The problem of control of the PMSM is actual, to support the intesestment of such problem many references can be given: (a) Torque Ripple Suppression of PMSM Based on Robust Two Degrees-of-Freedom Resonant Controller, (b) Critical Review on Robust Speed Control Techniques for Permanent Magnet Synchronous Motor (PMSM) Speed Regulation and (c) An adaptive proportional-integral-resonant controller for speed ripple suppression of PMSM drive due to current measurement error.
The manuscript is well prepared. The introduction is interesting. Only authors can add two newest references from 2023. This is because in the submitted manuscript authors cite one reference from 2023 and one references from 2023.
The manuscript linked strong mathematical model, focus on the robust control of PMSM with good experimental verification. In my opinion the minor improvements should be made by the authors. My remarks will be given in below list.
The detailed comments and remarks to the authors are following:
1. The authors propose the title of the manuscript "Combined Fractional-Order Repetitive Controller and Dynamic gain Regulator for Speed Ripple Suppression of PMSM Drive". In my opinion, word gain should be written in capital letter.
2. The authors in manuscript are focus on the FORC approach. In problem of control of power system currently are applied FOPDN controllers (Real-time validation of an automatic generation control system considering HPA-ISE with crow search algorithm optimized cascade FOPDN-FOPIDN controller). There is possible to adapted such type of controller in the control of PMSM?
3. In row 45 the authors wrote "Hall current sensors" in my opinion more correct is Hall effect current sensors (for discussion).
4. In Table 1 the authors presents parameters of PMSM. What mean parameters – Synchronous inductor. Please check correctness of writing this parameter.
5. In the Table 1 is given the reference speed ω, but in my opinion this is the angular velocity and unit should be [rad/s]. Please check correctness.
6. I suggest to write unit Rated torque TN [Nm] by using square brackets.
7. In row 113 the authors wrote reference [20] by superscript.
8. In many places on manuscript authors wrote [r·min-1] for angular velocity (row 122). In my opinion authors should change symbol on n and use [rpm] or use ω and [rad/s].
9. In row 141 the references ([21-22]) are written as a superscript.
10. In row 169 the authors introduce the symbol B (coefficient of friction?), but it is the lack of explains of this symbol.
11. In row 170 the authors wrote ierr(z) it is not consistent with Figure 4.
12. In the title of subsection the authors wrote title "Principle of FORC" in my opinion should be “of the FORC”.
13. Under equation (9) the k in the subscript are written by normal font, but in equation (9) the authors used the italic font. The writing of symbols should be unified in manuscript.
14. On Figure 13 the description of the x axis and numbers look as a e0.
15. The Figure 14 must be modified. At present look as a Figure from book.
16. In Table 2, authors use the normal letters as a subscripts (d, q) but in the description is italic font.
17. In row 233 the authors write "Its expressions can be given by:" and use the ":". In the manuscript there are introduction of equation with colon and without colon. The authors should unified manuscript.
18. Under equation (15) the authors wrote "Where" from capital letter. In my opinion authors should use normal letter.
19. In Figure 19, the authors present experimental results from different levels of speeds. It is possible to select a part for angular velocity equal 250, and apply magnification.

There are only minor modification in English language are needed (for example: Principle of FORC).
Author Response

(The authors gave the same response as above.)

Round 2
Reviewer 2 Report
Comments and Suggestions for Authors
Thank you for answering the question very clearly. I still have some doubts about the measurement error of the current sensor affecting the torque output ripple. The measurement error of the current sensor can be corrected during circuit design, and the noise can be removed using a digital filter. It must exist, and the control methods can be used to reduce torque output ripple.
However, I am delighted with your clear answers to all questions.